# Lie Detector: Unified Backdoor Detection via Cross-Examination Framework

**Xuan Wang**
National University of Defense Technology
wangxuan21d@nudt.edu.cn

**Siyuan Liang**[*]
National University of Singapore
pandaliang521@gmail.com

**Dongping Liao**
State Key Lab of IoTSC, CIS Dept,
University of Macau
yb97428@um.edu.mo

**Han Fang**
National University of Singapore
fanghan@nus.edu.sg

**Aishan Liu**
Beihang University
liuaishan@buaa.edu.cn

**Xiaochun Cao**
Sun Yat-sen University
caoxiaochun@mail.sysu.edu.cn

**Yuliang Lu**
National University of Defense Technology
publicLuYL@126.com

**Ee-chien Chang**[*]
National University of Singapore
changec@comp.nus.edu.sg

**Xitong Gao**[*]
Shenzhen Institutes of Advanced Technology, CAS
Shenzhen University of Advanced Technology
xt.gao@siat.ac.cn

## Abstract

Institutions with limited data and computing resources often outsource model training to third-party providers in an untrusted third-party setting, assuming adherence to prescribed training protocols with pre-defined learning paradigm (*e.g.,*supervised or self-supervised learning). However, this practice can introduce severe security risks, as adversaries may poison the training data to embed backdoors into the resulting model. Existing detection approaches predominantly rely on statistical analyses, which often fail to maintain universally accurate detection accuracy across different learning paradigms. To address this challenge, we propose a unified backdoor detection framework in the an untrusted third-party setting that exploits cross-examination of model inconsistencies between two independent service providers. Specifically, we integrate central kernel alignment to enable robust feature similarity measurements across different model architectures and learning paradigms, thereby facilitating precise recovery and identification of backdoor triggers. We further introduce backdoor fine-tuning sensitivity analysis to distinguish backdoor triggers from adversarial perturbations, substantially reducing false positives. Extensive experiments demonstrate that our method achieves superior detection performance, improving accuracy by 4.4%, 1.7%, and 10.6% over SoTA baselines across supervised, self-supervised, and autoregressive learning tasks, respectively. Notably, it is the first to effectively detect backdoors in multimodal large language models, highlighting its broad applicability and advancing secure deep learning.

## 1 Introduction

Deep learning models have grown exponentially in size in recent years, outstripping the computational resources available to many small and medium-sized institutions. Consequently, these institutions often rely on third-party cloud providers for model training. Although these providers are considered

---

[*]* Correspondence to Siyuan Liang, Ee-chien Chang and Xitong Gao.

39th Conference on Neural Information Processing Systems (NeurIPS 2025).

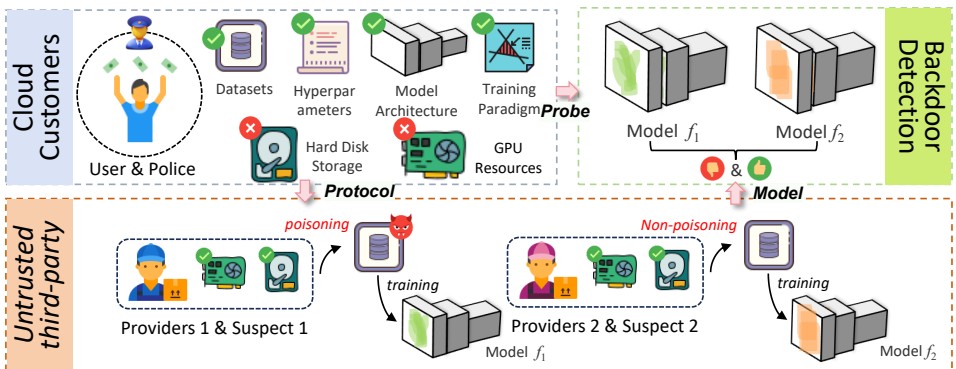

Figure 1: In the absence of training resources, the user delegates model training to a third-party vendor in an untrusted third-party environment and generates two independent models. At the same time, the user doubles as a police to identify potential backdoor models through comparative analysis.

"untrusted third-party" in that they ostensibly adhere to prescribed protocols, they may still covertly manipulate data or models. This scenario can give rise to a significant *backdoor threat*, where hidden triggers are embedded during training, enabling the model to function normally under most conditions but exhibit malicious behavior when specific triggers are activated [13, 34, 3, 25, 30, 28, 33, 29, 56].

Current backdoor detection methods frequently rely on model behavior and statistical analyses (*e.g.,*gradient-based detection, posterior analysis) [45, 35, 26, 12, 46, 47, 39]. Nevertheless, these methods exhibit significant sensitivity to fluctuations in optimization targets, loss functions, and feature representations across various learning paradigms [6]. This makes it hard for them to work with different architectures and attack tactics, which makes it hard to keep model security in untrusted third-party settings.

In many real-world scenarios, access to multiple independently trained models is feasible and practiced. Government, defense, and finance are examples of security-critical fields that commonly hire several outside companies to build models. This allows for cross-validation and builds trust. Modern machine learning pipelines, *e.g.,*federated learning, AutoML, and AutoTrain often train models at the same time with various seeds or settings. These realistic workflows naturally give rise to the multi-model input setting our method leverages.

Building on this observation, we propose the *Lie Detector* defense via a cross-examination backdoor detection framework designed for third-party verification. In Figure 1, the user (acting as `police`) gives the identical job to two different providers (the `suspects`) and finds backdoors by looking for differences in their model outputs (the `lies`). Specifically, we employ Central Kernel Alignment (CKA) [19, 4] for representation similarity, enabling the reverse-engineering of triggers (the `evidence`) by maximizing representational differences between two independently trained models. In contrast to conventional methods that rely on decision boundaries, our approach optimizes triggers based on output distributions, generalizing across supervised, self-supervised, and autoregressive learning paradigms. In order to reduce false positives and improve detection robustness, we also introduce a fine-tuning sensitivity analysis to distinguish between truly backdoored and benign models. With its consistent high detection accuracy across a variety of learning paradigms, Lie Detector provides a useful and adaptable backdoored model verification solution.

We thoroughly assess *Lie Detector*'s performance in supervised, self-supervised, and autoregressive settings. Our method works better than current methods, with relative increases of 4.4%, 1.7%, and 10.6% for SL, SSL, and AR, respectively. For instance, Lie Detector's 95% on COCO/TrojanVLM clearly outperforms previous techniques, which struggle on vision-language models (*e.g.,*LLaVA) with accuracies frequently below 50%. Furthermore, Lie Detector's robustness is highlighted by its great stability under a variety of random seeds. In order to increase security assurances for deep learning models, we believe that this research will promote a wider usage of secure training procedures in third-party services.

Our **contributions** are: **1)** We design a unified cross-examination framework for backdoor detection by analyzing inconsistencies in models provided by multiple third-party service providers, enhancing the security of outsourced training in untrusted third-party environments. **2)** Our method combines CKA for representation similarity and output distribution optimization, breaking the reliance on

decision boundaries and enabling backdoor detection to generalize beyond supervised learning to self-supervised learning and autoregressive learning. **3)** We achieve superior generalization across three learning paradigms and seven attack methods. Notably, it is the first to enable backdoor detection in multi-modal large language models, further broadening its applicability.

## 2 Related Work

**Development of Learning Paradigms.** Deep learning has evolved through training paradigms to address diverse challenges and data. This article focuses on supervised, self-supervised, and autoregressive learning, outlining their motivations, progress, and limitations. Supervised learning (SL) relies on labeled data, with early advances like CNNs [23] for image classification and DNNs for speech. Large-scale datasets (*e.g.,*ImageNet [7]) and architectures like ResNet [15] further propelled the field. However, dependence on costly annotations led to new paradigms. Self-supervised learning (SSL) mitigates this by generating supervisory signals from unlabeled data. BERT [8] revolutionized NLP, while SimCLR [2] and related methods advanced vision. Contrastive learning (CL), a subset of SSL, learns by distinguishing positive and negative pairs, with CLIP [40] and CoCoOp [53] demonstrating flexibility across modalities. Autoregressive learning (AL) builds on SSL and CL by modeling data distributions and generating samples, enabling cross-modal understanding via models like MiniGPT-4 [54] and LLaVA [24]. The evolution from SL to SSL and AL enhances adaptability and generalization, though high computational costs remain a barrier to broader adoption.

**Backdoor Attack.** Backdoor attacks have become a major security concern in deep learning, evolving alongside training paradigms. These attacks implant malicious behaviors during training, activated at inference by specific triggers. Early attacks focused on SL, leveraging labeled data to embed triggers. Examples include BadNets [13], which inserts poisoned data with fixed triggers; Blended [3], which blends patterns to evade detection; ISSBA [25], which uses invisible, sample-specific triggers for stealth; WaNet [38], which applies warping-based perturbations to boost success; and Low-Frequency [52], which exploits frequency features to implant hidden triggers. With the rise of SSL, attackers adapted or designed methods for unlabeled data. BadCLIP [31] compromises contrastive language-image models, while BadEncoder [18] poisons feature encoders to affect downstream tasks. As AL advances, attacks like TrojanVLM [27] embed multimodal triggers in vision-language models, and Shadowcast [51] targets text-to-image pipelines with stealthy backdoors. The attack surface now spans SL, SSL, and AL, with techniques tailored to each paradigm. This proliferation highlights the urgent need for robust, general-purpose defenses to safeguard models across tasks and modalities.

**Existing Backdoor Detection Methods.** Current methods often rely on model behavior and statistical analysis, such as gradients, activations, or output distributions [45, 35, 26, 12, 46, 37, 55]. Neural Cleanse (NC) [45] detects anomalies via trigger reversibility, while ABS [35] uses activation clustering to isolate poisoned neurons. NAD [26] applies knowledge distillation to suppress backdoors during fine-tuning. MM-BD [46] identifies arbitrary backdoor patterns via output landscape analysis and unsupervised anomaly detection. Some methods have extended detection to SSL and AL settings: DECREE [12] reverses triggers via optimization, and SEER [55] leverages auxiliary modalities for improved detection. While promising in specific paradigms, most of these methods lack scalability and fail to generalize across diverse learning settings. Our Lie Detector distinguishes itself in two key aspects: **1) Novelty.** We are the first to integrate CKA and fine-tuning sensitivity analysis for backdoor detection. While each has been used independently, their combination is purposefully designed to address both representation-level misalignment and behavioral instability, achieving complementary robustness. **2) Effect.** Our method is the first unified framework applicable to SL SSL, and AL. It also enables backdoor detection in large multimodal vision-language models (*e.g.,*LLaVA, MiniGPT-4), previously untouched by existing defenses, highlighting its broad applicability.

## 3 Preliminary

### 3.1 Threat Model

We consider a practical outsourced training scenario, where model training is performed by a third-party provider who may nominally follow the protocol but is essentially untrusted and may insert backdoors. The user has no visibility into the training process, but can independently inspect the returned model for malicious behavior.

**Adversarial capabilities**. During training or fine-tuning, the adversary, *e.g.*, a malicious service provider (the `suspects`), might introduce backdoors into the model by either directly altering the

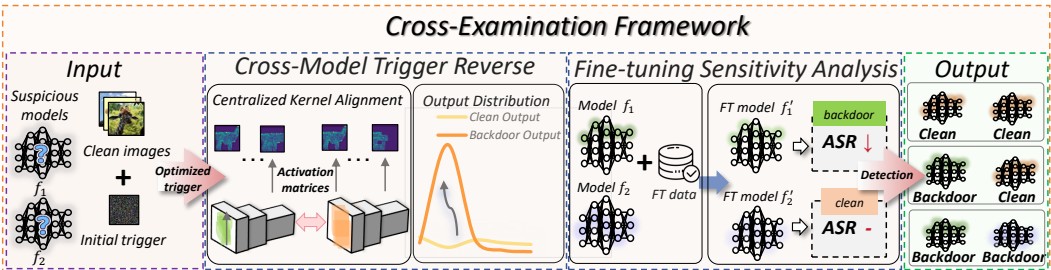

Figure 2: Overview of **Lie Detector**. We propose a general backdoor detection method via a cross-examination framework. Using output distribution and CKA losses to reverse triggers, and fine-tuning sensitivity analysis to identify backdoored models, our approach secures third-party training.

model's parameters to encode malicious behavior or by contaminating the training data with trigger patterns

**Adversarial knowledge**. The model architecture and training procedure may be completely known to the adversary, but they are not aware of the precise detection methods the verifier uses. This ensures that the backdoor detection framework remains robust against adaptive attacks.

**Detection constraints**. The verifier user (the `police`) cannot access the full training data or training process on the provider side but may have access to a small, trusted subset of clean data. There is no assumption of a clean reference model. This is consistent with real-world situations where the training process is treated as a "black box" and third parties are not privy to it.

## 3.2 Centered Kernel Alignment

CKA [49, 4, 19] measures the similarity between activations or feature representations. To compute CKA, we input data $\mathbf{X}$ into models $f^1$ and $f^2$ and extract activations from a specific layer $l$. Let $\mathbf{A}_1 \in \mathbb{R}^{n \times p_1}$ and $\mathbf{A}_2 \in \mathbb{R}^{n \times p_2}$ denote the extracted activation matrices of the two models, where $p_1$ and $p_2$ are the dimensionalities of the representations at that layer. The activations $\mathbf{A}_1$ and $\mathbf{A}_2$ are then transformed into kernel matrices $\mathbf{K}_1$ and $\mathbf{K}_2$ using a kernel function, typically a linear kernel:

$$\mathbf{K} = \mathbf{H}(\mathbf{A}\mathbf{A}^T)\mathbf{H}^\top, \ \mathbf{H} = \mathbf{I} - \frac{1}{n}\mathbf{1}\mathbf{1}^\top, \tag{1}$$

where $\mathbf{H}$ is the centering matrix, with $\mathbf{I}$ as the identity matrix and $\mathbf{1}$ a vector of ones. This transformation ensures that the kernel matrix $\mathbf{K} \in \mathbb{R}^{n \times n}$ eliminates biases due to differences in model architecture. The CKA similarity between the feature representations of two models is defined as:

$$\mathrm{CKA}(f^1, f^2, \mathbf{X}) = \frac{\langle \mathbf{K}_1, \mathbf{K}_2 \rangle_F}{\sqrt{\|\mathbf{K}_1\|_F^2 \cdot \|\mathbf{K}_2\|_F^2}}, \tag{2}$$

where $\langle \cdot, \cdot \rangle_F$ is the Frobenius inner product, and $\|\cdot\|_F^2$ the squared Frobenius norm, with $\langle \mathbf{K}_1, \mathbf{K}_2 \rangle_F = \mathrm{Tr}(\mathbf{K}_1^\top \mathbf{K}_2)$ and $\|\mathbf{K}_i\|_F^2 = \langle \mathbf{K}_i, \mathbf{K}_i \rangle_F$, where $\mathrm{Tr}(\cdot)$ is the matrix trace.

CKA is architecture independent because it remains unchanged under certain transformations [4, 19], meaning architectural differences do not affect model similarity. Specifically: 1) *Orthogonal transformation invariance*. CKA is unaffected by rotations or reflections of the feature space, making it robust to different basis representations. 2) *Isotropic scaling invariance*. Uniform scaling of features does not affect CKA, ensuring comparisons are unbiased by activation magnitudes. These properties make CKA ideal for comparing models with different architectures, as it captures the relative structure of representations rather than absolute values or network specifics.

## 4 Method

We introduce the **Lie Detector** via the cross-examination framework (Figure 2). Section 4.1 outlines the framework, while Section 4.2–4.4 detail the method and Algorithm 1 presents the full procedure.

### 4.1 Cross-Examination Framework

To secure third-party machine learning models, we propose a *Cross-Examination-Based Backdoor Detection Framework* under an *untrusted third-party verification setting*, comprising three modules:

**Cloud customers**. It consists of users who require model training services but lack direct control over the training process. These users also act as the verification party (the `police`), who have the

---

**Algorithm 1** Lie Detecor

---

**Require:** Models $f_1$, $f_2$; clean subset $\mathcal{D}_s$ & finetune set $\mathcal{D}_{ft}$; thresholds $\eta$, $\gamma$; weights $\alpha$, $\beta$, $\lambda$; epochs $T$; Adam
**Ensure:** Backdoor status of $f_1$ and $f_2$

1: Initialize trigger mask $\mathbf{m}$ and pattern $\mathbf{p}$       ▷ Stage I: Cross-Model Trigger Reverse
2: **for** $t = 1$ to $T$ **do**
3:     **for all** $(\mathbf{x}, y) \in \mathcal{D}_s$ **do**
4:       $\mathbf{x}' \leftarrow \mathbf{m} \odot \mathbf{p} + (1 - \mathbf{m}) \odot \mathbf{x}$       ▷ Generate poisoned input (Eq. (4))
5:       Compute $\mathcal{L}_{OD}$, $\mathcal{L}_{CKA}$       ▷ Eq. (5), Eq. (6)
6:     **end for**
7:     $\mathcal{L} \leftarrow \alpha \cdot \mathcal{L}_{CKA} + \beta \cdot \mathcal{L}_{OD} + \lambda \cdot (\|\mathbf{m}\|_1 + \|\mathbf{p}\|_1)$       ▷ Total loss (Eq. (7))
8:     Update the trigger $(\mathbf{m}, \mathbf{p})$ by minimizing $\mathcal{L}$ via Adam
9: **end for**
10: $\mathcal{D}_b \leftarrow \{\mathbf{x}' = \mathbf{m} \odot \mathbf{p} + (1-\mathbf{m}) \odot \mathbf{x} \mid \mathbf{x} \in \mathcal{D}_s\}$       ▷ Build poisoned set
11: **for** $f \in \{f_1, f_2\}$ **do**
12:     Predict $\tilde{y}_i = f(\mathbf{x}'_i)$, estimate target $\hat{y}_c$ by averaging       ▷ Stage II: Activation-Based Identification
13:     Compute $\text{ASR}(f) = \mathbb{E}[\mathbb{I}(f(\mathbf{x}') = \hat{y}_c)]$
14:     **if** $\text{ASR}(f) > \eta$ **then**       ▷ Stage III: Fine-tuning Sensitivity Analysis
15:       Fine-tune $f$ on $\mathcal{D}_{ft}$; compute $\text{ASR}(f')$; set $\Delta\text{ASR} \leftarrow \text{ASR}(f) - \text{ASR}(f')$
16:       **return Backdoored if** $\Delta\text{ASR} > \gamma$ **else return Clean**
17:     **else**
18:       **return Clean**
19:     **end if**
20: **end for**

---

authority to verify model integrity. The users provide the clean dataset $\mathcal{D}_c$, training hyperparameters, model architecture $f$, and the learning paradigm $\mathcal{L}_{learn}$.

**an untrusted third-party providers**. They are independent service providers (the `suspects`) responsible for model training. While following the user-specified protocol, they may still embed backdoors by poisoning part of the training data. Specifically, we assume the `suspect` constructs a poisoned training set by modifying a subset $\mathcal{D}_r \subset \mathcal{D}_c$ into $\mathcal{D}_p$, resulting in $\mathcal{D} = (\mathcal{D}_c \setminus \mathcal{D}_r) \cup \mathcal{D}_p$. The poisoned set $\mathcal{D}_p$ contains $\alpha|\mathcal{D}_c|$ samples, where $\alpha \in [0, 1]$ is the poison rate. The model $f_{\boldsymbol{\theta}}$ is then trained on $\mathcal{D}$. The adversary's learning process can be formulated as an optimization problem:

$$\arg\min_{\theta}\big\{\mathcal{L}_{\text{learn}}(f_\theta, \mathcal{D}) \triangleq (1 - \alpha)\mathbb{E}_{(\mathbf{x}_c, y_c) \sim \mathcal{D}_c}\big[\ell(f_\theta(\mathbf{x}_c), y_c)\big] + \alpha\mathbb{E}_{(\mathbf{x}_p, \hat{y}_c) \sim \mathcal{D}_p}\big[\ell(f_\theta(\mathbf{x}_p), \hat{y}_c)\big]\big\}, \quad (3)$$

where $y_c$ is the ground-truth label of a clean sample $\mathbf{x}_c$, and $\hat{y}_c$ the adversarial target for a poisoned sample $\mathbf{x}_p$. The objective $\mathcal{L}_{\text{learn}}$ depends on the paradigm, with loss function $\ell$ defined as: In *SL*, $y$ is a class label and $\ell$ is cross-entropy; in *CL* (*e.g.,*CLIP), $y$ encodes similarity and $\ell$ is contrastive; in *AL* (*e.g.,*LLaVA), $y$ is a reconstruction target and $\ell$ is autoregressive or reconstruction loss.

**Cross-examination backdoor detection**. The goal of cross-examination backdoor detection is to verify whether a model has been compromised without requiring access to its training data or process. Rather than relying on a known clean reference or predefined attack patterns, our approach detects backdoors by exploiting inconsistencies between two independently trained models ($f_1$ and $f_2$) from different third-party providers. Under this framework, there are three possible outcomes: *both models are clean, both models are backdoored, or one model is clean while the other is backdoored.*

Next, we outline the key challenges and motivations of our framework.

Challenges. Backdoor detection faces two key challenges: **1) Accuracy:** Many methods rely on statistical analysis and assume access to a clean reference model or known attack patterns. These assumptions often fail under unknown or adaptive attacks, as mismatched priors cause errors. Statistical methods also struggle to generalize beyond known attacks, limiting reliability. **2) Generalization:** A practical framework must generalize across architectures and learning paradigms, not just classification. Methods tied to specific models or objectives often fail in complex settings like self-supervised or generative learning. Architectural and task-agnostic generalization is essential for deployment.

Motivations. Our framework tackles these challenges with two key innovations: **1)** Exploiting model inconsistencies to avoid predefined attack priors. Traditional methods rely on assumptions about trigger or poison distributions, making them vulnerable to novel or adaptive attacks. We bypass this by comparing independently trained models on the same dataset, enabling detection without prior

attack knowledge. **2) Leveraging invariant features for better generalization.** Existing defenses often depend on specific architectures or tasks. In contrast, we target structural inconsistencies shared across models and paradigms, enabling broader applicability beyond classification.

## 4.2 Cross-Model Trigger Reverse

To identify potential backdoors, we reverse-engineer an effective trigger by exploiting behavioral discrepancies between two independently trained models $f_1$ and $f_2$. This stage serves as an initial screening for suspicious models. We randomly sample a subset $\mathcal{D}_s \subset \mathcal{D}_c$ of 1000 clean instances.

**Trigger formulation.** We adopt a trigger parameterization unified across various learning paradigms. Inspired by prior work on universal backdoor patterns [45], we define the trigger as a trainable pattern–mask pair. Let $\mathbf{p} \in \mathbb{R}^{H \times W \times C}$ be the injected pattern and $\mathbf{m} \in [0,1]^{H \times W \times C}$ the corresponding mask indicating modified pixels. For clean input $\mathbf{x}$, the poisoned input is defined as:

$$\mathbf{x}' = \mathbf{m} \odot \mathbf{p} + (1 - \mathbf{m}) \odot \mathbf{x}, \tag{4}$$

where $\odot$ is the element-wise product. By optimizing $\mathbf{m}$ and $\mathbf{p}$, we reconstruct effective triggers that are capable of eliciting malicious behavior in the suspect model.

**Output Distribution Loss.** To identify a backdoor trigger (`evidence`), we optimize the pair $(\mathbf{m}, \mathbf{p})$ defined in Eq. (4), so that the poisoned input $\mathbf{x}'$ induces abnormal behavior in the model. Backdoored models are typically trained to produce confident, target predictions that deviate from the ground-truth label, whereas clean models remain stable and preserve correct outputs under such perturbations. Minimizing the output distribution loss $\mathcal{L}_{\text{OD}}$ over clean samples guides the discovery of trigger patterns that accentuate this behavioral gap. The output distribution loss is defined as:

$$\mathcal{L}_{\text{OD}} = \frac{1}{|\mathcal{D}_s|} \sum_{(\mathbf{x}_i, y_i) \in \mathcal{D}_s} \begin{cases} -\ell_{\text{CE}}(f(\mathbf{x}_i'), y_i), & \text{if SL}, \\ \ell_{\text{Sim}}(f(\mathbf{x}_i'), f(y_i)), & \text{if SSL}, \\ -\sum_t \ell_{\text{AR}}(f(\mathbf{x}_i')^{(t)}, y_i^{(t)}), & \text{if AL}. \end{cases} \tag{5}$$

For SL, $\ell_{\text{CE}}$ is the cross-entropy loss w.r.t. the ground-truth label $y_i$; minimizing its negative encourages confident misclassification. For SSL, $\ell_{\text{Sim}}$ penalizes similarity between poisoned and clean features, with lower values indicating semantic drift. For AL, $\ell_{\text{AR}}$ measures generation error against the ground-truth $y_i$; minimizing its negative reveals sequence-level deviations.

**CKA-based loss.** To capture internal discrepancies caused by backdoors, we leverage CKA as a representation-level metric. Unlike output-based measures, CKA quantifies alignment between intermediate feature spaces, making it suitable for comparing models across different architectures or objectives. By maximizing this divergence, we highlight the `Lie` hidden within a suspect model. Let $\mathbf{K}_1^l, \mathbf{K}_2^l$ denote kernel matrices from activations at layer $l$ of models $f_1$ and $f_2$ given input $\mathbf{x}'$. The CKA loss is defined as:

$$\mathcal{L}_{\text{CKA}}(\mathbf{x}', f_1, f_2, l) = \frac{\langle \mathbf{K}_1^l(\mathbf{x}'), \mathbf{K}_2^l(\mathbf{x}') \rangle_F}{\sqrt{\|\mathbf{K}_1^l(\mathbf{x}')\|_F^2 \cdot \|\mathbf{K}_2^l(\mathbf{x}')\|_F^2}}. \tag{6}$$

**Joint objective.** The full optimization combines the output-level and representation-level signals:

$$\mathcal{L}(\mathbf{m}, \mathbf{p}) = \alpha \cdot \mathcal{L}_{\text{CKA}} + \beta \cdot \mathcal{L}_{\text{OD}} + \lambda \cdot (\|\mathbf{m}\|_1 + \|\mathbf{p}\|_1), \tag{7}$$

where the $\|\cdot\|_1$ regularization promotes sparsity and limits perturbation magnitude, thereby aiding trigger optimization, following the design principles of DECREE [12].

## 4.3 Activation-Based Identification via Trigger-Induced ASR

We introduce two post-trigger criteria to determine whether a model is backdoored. The first criterion evaluates the model's behavior under trigger injection, serving as a fast and effective filter.

The motivation lies in the fact that backdoored models are explicitly trained to associate specific trigger patterns with an attacker-defined target label, resulting in consistent and confident mispredictions when the trigger is applied. In contrast, clean models lack such associations and typically retain correct predictions under the same perturbations. Given the poisoned set $\mathcal{D}_b = \{\mathbf{x}_i'\}$ constructed from clean inputs $\mathcal{D}_s$ using Eq. (4), we compute the prediction $\tilde{y}_i = f(\mathbf{x}_i')$ for each model $f$, and estimate the target label $\hat{y}_c$ by averaging over all $\tilde{y}_i$. The attack success rate (ASR) is then defined as:

$$\text{ASR}(f) = \mathbb{E}_{\mathbf{x}' \in \mathcal{D}_b} \left[ \mathbb{I}(f(\mathbf{x}') = \hat{y}_c) \right]. \tag{8}$$

A model is flagged as backdoored if:

$$\text{Backdoored} \iff \text{ASR}(f) > \eta. \tag{9}$$

This criterion offers a straightforward and effective filter for strongly triggered behaviors, but may suffer from false positives in certain edge cases.

### 4.4 Fine-tuning Sensitivity Analysis for Robust Backdoor Confirmation

To mitigate the limitations of Criterion I, we propose a robustness-based test: fine-tune the model on a small clean subset $\mathcal{D}_{\text{ft}} \subset \mathcal{D}_c$ (*e.g.,*10%) using its original objective:

$$f' = \arg\min_f \mathcal{L}_{\text{learn}}(f, \mathcal{D}_{\text{ft}}). \tag{10}$$

The motivation is that clean models naturally generalize well to clean data, and their predictions remain stable under further fine-tuning. In contrast, backdoored models embed spurious dependencies on trigger patterns; fine-tuning on clean samples disrupts these dependencies and deactivates the backdoor effect. We then recompute the ASR on the fine-tuned model $f'$:

$$\text{ASR}(f') = \mathbb{E}_{\mathbf{x}' \in \mathcal{D}_b} \left[ \mathbb{I}(f'(\mathbf{x}') = \hat{y}_c) \right]. \tag{11}$$

A significant drop in ASR indicates a disrupted backdoor and confirms the malicious dependency:

$$\text{Backdoored} \iff \Delta\text{ASR} = \text{ASR}(f) - \text{ASR}(f') > \gamma. \tag{12}$$

*Together, these two criteria form a dual-phase verification pipeline: the first captures activation, and the second verifies its robustness. Criterion II also represents a key innovation of our framework, enabling accurate identification with reduced false positives.*

## 5 Experiments

### 5.1 Implementation Details

**Models and Datasets.** We evaluate across diverse learning paradigms. For *supervised learning*, we use ResNet18 [15] and VGG16 [42] on CIFAR-10 [21] and TinyImageNet [44]. For *self-supervised and autoregressive learning*, we test CLIP [40] and CoCoOp on ImageNet [7] and Caltech101 [11], while LLaVA [24] and MiniGPT-4 [54] are evaluated on COCO [32], Frisk-30k [10], and Frisk-8k [16].

**Attacks and Defenses.** We consider backdoor attacks across different paradigms, including Bad-Nets [13], Blended [3], ISSBA [25], WaNet [38], and Low-Frequency [52] for *supervised learning*. For *self-supervised and autoregressive learning*, we adapt these attacks and further evaluate Bad-CLIP [31], BadEncoder [18], TrojanVLM [27], and Shadowcast [51]. We employ advanced defenses, including NC [45], ABS [35], NAD [26], UNICORN [48], MM-BD [46], DECREE [12], and SEER [55]. Some methods, such as MM-BD, are extended to multiple paradigms. Unless otherwise specified, all attacks use a 10% poison rate. All evaluations are conducted using the untrusted third-party environment, with detailed settings and evaluation metrics in Appendices B.1 and B.2.

**Configurations.** In our experiments, we use equal numbers of clean and backdoored models. In each evaluation, two models are randomly sampled to form clean–clean, clean–backdoored, or backdoored–backdoored pairs. To ensure a fair comparison, we randomly select 20 model pairs (without repetition) for testing and compute the detection performance by averaging their scores. The trigger is optimized with Adam. We use default hyperparameters for Algorithm 1: $\gamma = 0.2$, $\eta = 0.75$, $\alpha = 0.6$, $\beta = 0.3$, $\lambda = 0.1$, $T = 100$.

**Evaluation Assumption.** All comparisons between suspicious models and defense methods in this work are conducted under the assumption that the evaluator (or defender) has access to at least one clean model, which serves as a reference for comparison and judgment. This is consistent with the settings adopted in prior studies [50, 37]. In cases where this assumption does not hold, the behaviors and threshold choices of many defense methods may differ substantially in a fully black-box scenario without any clean baseline.

### 5.2 Detection Performance in SL

In Table 1, we compare our **Lie Detector** with 7 SOTA detection methods against 4 representative attacks on CIFAR-10 and TinyImageNet. **Lie Detector consistently achieves 100.0% DSR across all attacks and datasets**, with average gains of **4.4%** over the next-best methods and near-zero FPR. In contrast, others often fail under adaptive or stealthy attacks, particularly ISSBA and Low-Frequency. On CIFAR-10, Lie Detector surpasses the next-best method by **5.0%** on ISSBA and **7.5%** on Low-Frequency. On TinyImageNet, similar margins appear: **5.0%** on Blended and **7.5%** on Low-Frequency. These gains come with consistently lower FPR (0.0–5.0%) compared to higher rates from other methods. While recent defenses like UNICORN, MM-BD, and DECRE improve over early detectors (*e.g.,*NC, ABS, NAD), they still lack robustness under challenging attacks. Notably, while both NC and Lie Detector perform trigger reversal in the spatial domain, our method introduces

Table 1: Detection performance (%) on SL (ResNet-18). For each attack, we evaluate 20 clean and 20 backdoored models. Detection Success Rate (DSR) and False Positive Rate (FPR) are reported.

| Dataset | Attack | NC [45] | | ABS [35] | | NAD [26] | | UNICORN [48] | | MM-BD [46] | | DECREE [12] | | Lie Detector | |
|---|---|---|---|---|---|---|---|---|---|---|---|---|---|---|---|
| | | DSR | FPR | DSR | FPR | DSR | FPR | DSR | FPR | DSR | FPR | DSR | FPR | DSR | FPR |
| CIFAR10 | BadNet | 87.5 | 10.0 | 90.0 | 5.0 | 92.5 | 15.0 | **100.0** | 5.0 | **100.0** | **0.0** | 97.5 | **0.0** | **100.0** | **0.0** |
| | Blended | 30.0 | 25.0 | 80.0 | 15.0 | 67.5 | 10.0 | 92.5 | 5.0 | 97.5 | **0.0** | 92.5 | 5.0 | **100.0** | **0.0** |
| | ISSBA | 25.0 | 30.0 | 37.5 | 40.0 | 50.0 | 20.0 | 90.0 | 5.0 | 92.5 | 5.0 | 90.0 | 5.0 | **100.0** | **0.0** |
| | Low-Freq. | 10.0 | 55.0 | 25.0 | 45.0 | 20.0 | 30.0 | 60.0 | 25.0 | 92.5 | 5.0 | 87.5 | 10.0 | **100.0** | **0.0** |
| TinyImgNet | BadNet | 77.5 | 10.0 | 80.0 | 10.0 | 82.5 | 20.0 | 90.0 | 5.0 | 95.0 | **0.0** | 97.5 | **0.0** | **100.0** | **0.0** |
| | Blended | 15.0 | 30.0 | 70.0 | 20.0 | 42.5 | 15.0 | 90.0 | 10.0 | 92.5 | 5.0 | 95.0 | **0.0** | **100.0** | **0.0** |
| | ISSBA | 10.0 | 40.0 | 25.0 | 25.0 | 40.0 | 25.0 | 85.0 | 10.0 | 95.0 | 5.0 | 92.5 | 10.0 | 97.5 | 5.0 |
| | Low-Freq. | 5.0 | 50.0 | 12.5 | 45.0 | 30.0 | 30.0 | 45.0 | 35.0 | 92.5 | 5.0 | 90.0 | 5.0 | **100.0** | 5.0 |

Table 2: Detection performance (%) on SSL (CLIP) and AL (LLaVA). We evaluate 20 clean and 20 backdoored models per attack. DSR and FPR are reported.

| Architecture | Dataset | Attack | MM-BD [46] | | DECREE [12] | | SEER [55] | | Lie Detector | |
|---|---|---|---|---|---|---|---|---|---|---|
| | | | DSR | FPR | DSR | FPR | DSR | FPR | DSR | FPR |
| CLIP | Caltech101 | BadNet | 75.0 | 10.0 | 87.5 | 20.0 | **100.0** | **0.0** | **100.0** | **0.0** |
| | | Blended | 72.5 | 20.0 | 82.5 | 25.0 | **97.5** | **0.0** | **97.5** | **0.0** |
| | | BadCLIP | 52.5 | 25.0 | 60.0 | 30.0 | 90.0 | **0.0** | **95.0** | 5.0 |
| | ImageNet | BadNet | 67.5 | 10.0 | 72.5 | 10.0 | **95.0** | **0.0** | **95.0** | **0.0** |
| | | Blended | 65.0 | 15.0 | 75.0 | 15.0 | 90.0 | 5.0 | **92.5** | **0.0** |
| | | BadCLIP | 42.5 | 30.0 | 45.0 | 20.0 | 87.5 | 10.0 | **90.0** | 5.0 |
| LLaVA | COCO | TrojanVLM | 15.0 | 45.0 | 60.0 | 40.0 | 80.0 | 15.0 | **95.0** | 5.0 |
| | | Shadowcast | 15.0 | 50.0 | 60.0 | 45.0 | 85.0 | 10.0 | **92.5** | **0.0** |
| | Flickr-30K | TrojanVLM | 15.0 | 50.0 | 55.0 | 45.0 | 80.0 | 5.0 | **90.0** | 10.0 |
| | | Shadowcast | 10.0 | 45.0 | 45.0 | 35.0 | 80.0 | 10.0 | **90.0** | 5.0 |

CKA to measure representational misalignment between models. This enables detection of semantic inconsistencies even when trigger patterns are smooth or imperceptible, as in Low-Frequency attacks. Even UNICORN, which considers feature- and frequency-space triggers, underperforms in such cases, highlighting the strength of CKA-driven alignment in exposing cross-model backdoor discrepancies.

## 5.3 Detection Performance in SSL and AL

We evaluate our method against 4 defenses under SSL and AL paradigms, with default implementations and modest modifications, across 4 datasets and 3 attacks. Table 2 concludes: 1) **Limited generalization.** Traditional methods (MM-BD, DECREE) show inconsistent performance across datasets and architectures. While some perform well on CLIP (*e.g.,*DECREE: 87.5% DSR on Caltech101/BadNet), they fail on vision-language models (*e.g.,*LLaVA), with DSRs often below 50% (*e.g.,*MM-BD: 15%, DECREE: 60% on COCO/TrojanVLM), indicating weak adaptability. 2) **High FPR.** Many methods, particularly MM-BD, exhibit FPRs up to 50%, misclassifying clean models as backdoored with detection rates no better than random guessing. In contrast, our method shows better generalization, maintaining high DSR (*e.g.,***95.0%** on COCO/TrojanVLM, **90.0%** on Flickr/Shadowcast) and low FPR ($\leq$**10.0%**), with average gains of **1.7%** for SSL and **10.6%** for AR over the next-best methods.

## 5.4 Adaptive Attack on Lie Detector

To further validate the robustness of the proposed method, we conduct adaptive attack experiments, which simulate adaptive attackers by modifying the training process of the backdoor model.

We design an adaptive attacker that is aware of our detection and aims to bypass our CKA-based representation divergence detector. In particular, the attacker first trains a clean reference model $f_{\text{clean}}$ on the clean training dataset. Then, when training the adaptive backdoored model $f$ on the poisoned training dataset, the attacker combines the original training loss $\mathcal{L}_{\text{origin}}$ with a regularization term designed to suppress detection. This regularizer maximizes the CKA similarity between the clean and backdoored models:

$$\mathcal{L}_{\text{adaptive}} = \mathcal{L}_{\text{origin}} - \lambda \cdot \mathcal{L}_{\text{CKA}}(f, f_{\text{clean}}) \tag{13}$$

$\mathcal{L}_{\text{CKA}}$ denotes the CKA similarity between intermediate representations of the two models under the same input as used in our paper, and $\lambda$ controls the strength. We vary $\lambda$ and report the attacker's ASR as well as our defense's DSR and FPR in the table below.

In Table 3, as $\lambda$ increases the attacker can partly suppress cross-model representation divergence, causing DSR to drop from 99.61% to 86.54%; ASR also falls from 98.75% to 82.37%, indicating a

Table 3: Performance under adaptive attacks with varying $\lambda$.

| $\lambda$ | ASR(%) | DSR(%) | FPR(%) |
|---|---|---|---|
| 0 (No adaptive) | 98.75 | 100.0 | 0.0 |
| 0.1 | 94.90 | 92.5 | 0.0 |
| 0.2 | 90.68 | 90.0 | 5.0 |
| 0.5 | 82.37 | 85.0 | 10.0 |

Table 4: Component ablation experiments. (FTSA = Fine-tuning Sensitivity Analysis; CKA = Centered Kernel Alignment loss)

| Attack | Task | Trigger Size | Model | wo/ FTSA | | wo/ CKA | | Lie Detector (Ours) | |
|---|---|---|---|---|---|---|---|---|---|
| | | | | DSR | FPR | DSR | FPR | DSR | FPR |
| Blended | CIFAR10 | 4×4 | ResNet-18 | 100.0 | 10.0 | 85.0 | 7.5 | 100.0 | 0.0 |
| | | | VGG16 | 100.0 | 20.0 | 82.5 | 10.0 | 100.0 | 0.0 |
| BadEncoder | Caltech101 | 32×32 | CLIP | 100.0 | 20.0 | 80.0 | 10.0 | 95.0 | 2.5 |
| | | | CoCoOp | 100.0 | 20.0 | 82.5 | 5.0 | 100.0 | 0.0 |
| Shadowcast | Flickr8k | 50×50 | LLaVA | 90.0 | 20.0 | 77.5 | 10.0 | 92.5 | 2.5 |
| | | | Mini-GPT4 | 90.0 | 30.0 | 75.0 | 7.5 | 90.0 | 0.0 |

weakened backdoor. This demonstrates a trade-off between stealth and efficacy, even under adaptive attacks our method raises the difficulty and cost of stealthy backdoor injection.

### 5.5 Discussion

**Component ablation.** To validate the effectiveness of **Lie Detector** under different configurations. Specifically, with or without "Fine-tuning Sensitivity Analysis (FTSA)" and the "Centered Kernel Alignment (CKA) loss", we conduct component ablation experiments in Table 4. The results lead to three conclusions: (1) the "Cross-Model Trigger Reverse" method combined with basic "Activation-Based Identification" is effective but less robust, while achieving high Detection Success Rates (DSR), it suffers from elevated False Positive Rates (FPR), reaching up to 30% in some cases, indicating that some clean models are misclassified as backdoored; (2) introducing "Fine-tuning Sensitivity Analysis" significantly enhances robustness, reducing FPR to 0% across all settings while maintaining equally high DSR, effectively distinguishing backdoored models from clean ones; and (3) removing the *CKA loss term* (Eq. (7)) leads to noticeable degradation in performance. As shown in A5, eliminating CKA results in clear drops in DSR (e.g., $100.0 \rightarrow 85.0$ on CIFAR-10/Blended) and increases in FPR (e.g., $0.0 \rightarrow 7.5$), confirming that the CKA loss provides complementary representation-level supervision during trigger optimization. This is particularly crucial for complex architectures (e.g., CLIP, LLaVA), where relying solely on output distributions may fail to sufficiently expose backdoor inconsistencies, while CKA effectively compensates for this limitation.

**Number of epochs $T$.** We show the detection accuracies (DSR) as the number of epochs increases for ResNet-18 and CLIP models in Figure 3. We observe that our method converges stably and remains effective across all attack methods on both ResNet-18 and CLIP. The DSR consistently improves with training epochs, demonstrating the robustness and adaptability of our approach in identifying backdoored models across learning paradigms.

**Feature layer selection.** As shown in Table 6, CKA values in clean models remain stable across layers, while backdoored models show a clear decline in deeper layers. This trend holds for both ResNet-18 (SL) and CLIP (SSL), confirming CKA's reliability as a backdoor probe. Notably, layer 4 yields the largest CKA drop (0.427 in ResNet-18, 0.314 in CLIP), making it the most effective for detection. A likely reason is that deeper-layer features capture more abstract semantics, which backdoor triggers distort, leading to greater representation shifts. We therefore use fourth-layer features to compute the CKA loss. Further analysis of CKA under varying poison rates and across clean/backdoored models is provided in Appendix A.

**Model architecture**. We evaluate our detection method across six model architectures spanning three learning paradigms, as shown in Table 7. Specifically, we assess supervised models (ResNet-18, VGG16), contrastive language-image models (CLIP, CoCoOp), and vision-language models (LLaVA, Mini-GPT4) under three attacks. Results show: (1) Our method achieves 100% DSR and 0% FPR across all architectures, including complex multimodal models like LLaVA and Mini-GPT4. (2) Detection remains unaffected by model complexity, as measured by FLOPs. For instance, despite the

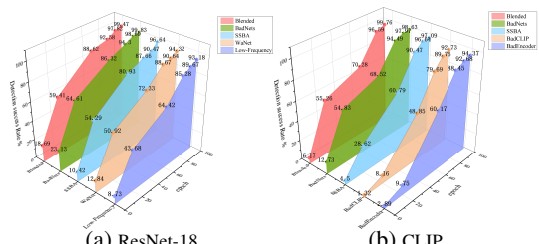

(a) ResNet-18  (b) CLIP

Figure 3: DSR as the number of epochs $T$.

Table 6: CKA value variation across layers.

| Layer | ResNet-18 | | CLIP | |
|---|---|---|---|---|
| | Clean | Backdoor | Clean | Backdoor |
| 1 | 0.974 | 0.945 | 0.891 | 0.863 |
| 2 | 0.936 | 0.768 | 0.853 | 0.632 |
| 3 | 0.901 | 0.542 | 0.810 | 0.497 |
| 4 | 0.872 | **0.427** | 0.795 | **0.314** |

Table 7: Performance across model architectures.

| Attack | Task | Trigger Size | Model | DSR | FPR | GFLOPs |
|---|---|---|---|---|---|---|
| BadNet | CIFAR10 | 4×4 | ResNet-18 | 100.0 | 0.0 | 0.7 |
| | | | VGG16 | 100.0 | 0.0 | 0.4 |
| BadCLIP | Caltech101 | 32×32 | CLIP | 90.0 | 0.0 | 4.9 |
| | | | CoCoOp | 100.0 | 0.0 | 5.0 |
| TrojanVLM | Flickr8k | 50×50 | LLaVA | 90.0 | 0.0 | 76.6 |
| | | | Mini-GPT4 | 80.0 | 0.0 | 80.3 |

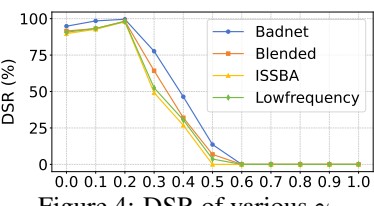

Figure 4: DSR of various $\gamma$.

increase in computational cost from ResNet-18 (0.7 GFLOPs) to Mini-GPT4 (80.3 GFLOPs), our method maintains perfect detection with zero false positives.

**Hyperparameter $\gamma$.** As shown in Figure 4, we evaluate the impact of $\gamma$ on detection accuracy across multiple backdoor attacks. When $\gamma = 0$, the framework relies solely on the activation-based criterion and achieves high detection accuracy. However, as $\gamma$ increases beyond 0.2, performance drops sharply. At $\gamma = 0.6$, all attacks converge to 0% detection accuracy, indicating that overly strict reliance on the ASR-drop criterion may suppress correct detections.

**Impact of similarity metric and poison rate.** We evaluate four metrics (CKA, CCA, SVCCA, and COS [22]) under five poison rates (0.1%, 1%, 5%, 10%, 20%) against the Blended attack. CKA consistently outperforms others across learning paradigms, especially on LLaVA, reaching an F1 score of **0.92** at 10% poison rate, and vs. 0.55 (COS), 0.50 (CCA), and 0.47 (SVCCA), demonstrating robustness to subtle backdoors and model-agnostic performance. While detection improves with higher poison rates, traditional metrics degrade under low poisoning; for example, at 1% on CLIP, CKA achieves **0.50** while others remain below 0.3. Full results are in Appendix D.

**Comparison with existing methods.** Appendix F shows that Lie Detector offers lower cost than NAD, and works without label supervision, relying only on clean data. It supports SL, SSL, and AL, unlike DECRE (pre-training only) and others limited to classification. While it requires two independently trained models, this is practical in many real-world settings. Overall, it achieves the highest detection success rate (99.7%), outperforming all baselines.

**Potential advanced scenarios.** We outline three advanced cases (details in Appendix E): 1) *Collusion Attacks.* Multiple providers may collude to train aligned backdoored models, concealing discrepancies and bypassing cross-model detection. 2) *Client-level Inspection in FL.* In a simplified FL setup, clients train separately and submit models for centralized auditing, akin to detecting poisoned updates before aggregation. 3) *Scaling to Larger Models.* Extending our approach to large-scale architectures.

## 6 Conclusion

This paper proposes Lie Detector, a unified backdoor detection framework for untrusted third-party settings where model training is outsourced to third-party providers. By leveraging cross-examination of inconsistencies between independent providers, our method significantly improves detection robustness across learning paradigms. We integrate Centered Kernel Alignment (CKA) for precise feature similarity measurement and fine-tuning sensitivity analysis to distinguish backdoor triggers from adversarial perturbations, effectively reducing false positives. Extensive experiments show that our approach outperforms state-of-the-art methods, achieving superior detection accuracy in supervised, contrastive, and autoregressive tasks. Notably, it is the first to effectively detect backdoors in multimodal large language models. This work offers a practical solution to mitigate backdoor risks in outsourced training, paving the way for more secure and trustworthy AI systems. We also provide the limitations of our work in Appendix I, discussing *generality and scalability* as future directions.

## Acknowledgments

This work is supported in part by Yu Liang Lu's Project Team Development Funding (KY23A102), National Natural Science Foundation of China (62376263), Natural Science Foundation of Guangdong (2024A1515030209), and Shenzhen Science and Technology Innovation Commission (JCYJ20230807140507015).

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

# A    CKA Effectiveness Analysis

To validate that CKA (Centered Kernel Alignment) effectively highlights differences between models and can distinguish clean models from backdoored ones, we select a specific poisoned dataset to test whether CKA values differ between two clean models versus those involving a backdoored model. This experiment aims to verify the discriminative capability of CKA in detecting backdoor attacks.

Table Appendix 1: Comparison of CKA under different poison rates

| poisoned data(poison rate=0.1) | | |
|---|---|---|
| Model 1 | Model 2 | Similarity |
| clean_model_1 | clean_model_2 | 0.801 |
| backdoor_model_1 | backdoor_model_2 | 0.311 |
| backdoor_model_1 | backdoor_model_3 | 0.385 |
| clean_model_1 | backdoor_model_1 | 0.254 |
| clean_model_2 | backdoor_model_1 | 0.249 |
| clean_model_1 | backdoor_model_2 | 0.267 |
| clean_model_2 | backdoor_model_2 | 0.253 |
| clean_model_1 | backdoor_model_3 | 0.368 |
| clean_model_2 | backdoor_model_3 | 0.316 |
| **poisoned data (poison rate=0.2)** | | |
| clean_model_1 | clean_model_2 | 0.801 |
| backdoor_model_1 | backdoor_model_2 | 0.286 |
| backdoor_model_1 | backdoor_model_3 | 0.391 |
| clean_model_1 | backdoor_model_1 | 0.263 |
| clean_model_2 | backdoor_model_1 | 0.236 |
| clean_model_1 | backdoor_model_2 | 0.229 |
| clean_model_2 | backdoor_model_2 | 0.214 |
| clean_model_1 | backdoor_model_3 | 0.351 |
| clean_model_2 | backdoor_model_3 | 0.325 |

In table Appendix 1, we can see that there is a backdoor model, the CKA between the two models will be much lower than the CKA between two clean models, and this phenomenon is robust to changes in the poisoned rate. Additionally, we tested the trigger inversion capability of CKA on the CLIP.

We used the attack success rate as the metric to evaluate the capability of reverse trigger detection. Experimental results demonstrate that, compared to the four existing similarity measurement methods, our approach achieves the best performance in trigger inversion.

# B    Additional Details

## B.1    Attack Setting

**Attack parameters**. Unless otherwise specified, all attack methods are configured with a 10% poison rate, meaning 10% of the training data is poisoned to simulate real-world adversarial conditions. The number of backdoor training images used for poisoning was carefully chosen for each backdoor pattern and for each dataset to ensure a high attack success rate for the created backdoor attacks. Details are shown in Tab. Appendix 2.

The detailed implementations for all backdoor attack methods are given below:

**BadNets** [13]. We follow the attack methodology proposed by BadNets, and this work belongs to the simple backdoor attack. Backdoor injection during training, we inject adversarial inputs by randomly selecting a target label and modifying the training data. The adversarial input is created by applying a trigger, which a white square in the bottom right corner of the image that does not cover any significant part, such as faces or symbols. The trigger's shape and color are chosen to ensure uniqueness and to prevent it from occurring naturally in the input images. To keep the trigger subtle, its size is limited to about 1% of the entire image.

Table Appendix 2: Training Configuration for Different Datasets and Models

| Parameter | CIFAR-10 | TinyImagenet | Caltech101 | COCO |
|---|---|---|---|---|
| Model | ResNet-18 | VGG-16 | CLIP | VLM |
| Optimizer | Adam | Adam | Adam | Adam |
| Batch Size | 64 | 128 | 224 | 224 |
| Epochs | 60 | 100 | 100 | 100 |
| Image Size | $32 \times 32$ | $64 \times 64$ | $224 \times 224$ | $224 \times 224$ |
| Learning Rate | $1 \times 10^{-3}$ | $1 \times 10^{-4}$ | $1 \times 10^{-3}$ | $1 \times 10^{-3}$ |

**Blended** [3]. We follow the attack methodology proposed by Blended and treat it as a simple backdoor attack. Backdoor injection is performed during training by overlaying a global trigger, typically a fixed pattern such as a translucent image onto the entire input image. The trigger is blended with the original image using a low opacity (e.g., blending ratio of 0.2) to ensure that it is visually unobtrusive. The target label is fixed and used for all poisoned samples. The trigger pattern is designed to be unique and unlikely to appear in natural images, ensuring its effectiveness during inference.

**ISSBA** [25]. We directly use the ISSBA backdoor attack method in the original paper. This method belongs to the specific label attack. This method employs an encoder-decoder network to embed a string specified by the attacker into a benign image as the backdoor pattern. The encoder constructs the poisoned image, aiming to minimize the difference between the poisoned and normal images. The decoder decodes the triggers in the poisoned image, minimizing the reconstruction loss of the encoding.

**Low-Frequency** [52]. We follow the attack methodology in the original paper and consider it as a spectral-domain backdoor attack. During training, poisoned samples are generated by adding adversarial perturbations constrained to the low-frequency components of the input image. This is achieved via Discrete Cosine Transform (DCT), where perturbations are restricted to low-frequency subbands. These perturbations are imperceptible to human eyes but can significantly degrade model generalization. A fixed target label is assigned to all poisoned examples to enable the backdoor effect during inference.

**WaNet** [38]. We follow the attack methodology in the original paper, which is a warping-based clean-label backdoor attack. During training, we apply a subtle image-warping operation to a subset of training samples using a smooth and learnable warping field, while keeping their labels unchanged. The warping field is constructed from a randomly generated control point grid passed through a thin-plate spline transformation, ensuring natural-looking distortions. At test time, a fixed warping trigger is applied to activate the backdoor. The trigger is designed to be imperceptible to humans, making the poisoned inputs visually indistinguishable from clean data.

**BadCLIP** [31]. For the implementation of BadCLIP, we follow the methodology in the original paper. BadCLIP is a backdoor attack targeting multimodal contrastive learning models such as CLIP. During pretraining, a small set of image-text pairs is poisoned by inserting a visual trigger into the image and aligning it with a fixed target text prompt. The dual-embedding optimization encourages the poisoned samples to be pulled toward the target prompt in the joint embedding space while preserving performance on clean samples. The visual trigger is small and imperceptible, ensuring stealthiness and effectiveness.

**BadEncoder** [18]. We follow the official implementation of BadEncoder, which introduces a backdoor into the visual encoder of multimodal models. A learnable perturbation is added to all input images during training to construct a universal adversarial feature space. The poisoned visual encoder is optimized to align these features with a target prompt, enabling targeted manipulation at test time. The attack is clean-label and does not require modifying the textual input.

**TrojanVLM** [27]. We implement TrojanVLM by following the official training pipeline. This attack injects backdoors into large pre-trained vision-language models through prompt-based tuning. A trigger prompt (e.g., a specific phrase or token) is injected into the textual input, and clean images are used during training. The attack encourages the model to misinterpret benign visual content as matching the target class when the trigger phrase appears in the prompt. The visual encoder remains fixed while tuning the textual components.

**ShadowCast** [51]. For ShadowCast, we follow the official implementation, which constructs unlearnable examples by injecting stealthy adversarial perturbations into both the visual and textual modalities of vision-language models. During training, perturbations are optimized to reduce the model's ability to learn meaningful alignment between image-text pairs, without affecting human perception. The resulting poisoned dataset causes a significant degradation in downstream performance while preserving data utility for human observers.

## B.2 Defense Setting

**Detection protocol.** We evaluate each detection method under an untrusted third-party environment where only limited clean data is available for verification. Specifically, each dataset is split into a 90%-10% training-validation ratio, with only 10% clean data accessible for detection. We report two key metrics: Detection Success Rate (DSR), which measures the percentage of correctly identified backdoored models, and False Positive Rate (FPR), which quantifies the rate of clean models misclassified as backdoored.

**Evaluation across learning paradigms.** To demonstrate the generalizability of our method, we test it across different learning paradigms. For supervised learning, we use ResNet18 and VGG16 trained on CIFAR-10 and TinyImageNet. For self-supervised learning, we evaluate CLIP and CoCoOp on ImageNet and Caltech101. For autoregressive learning, we test LLaVA and Mini-GPT4 on COCO and Flickr-30k.

**Implementation details.** All experiments are conducted using PyTorch, with models trained on NVIDIA A100 GPUs. For fair comparison, we fine-tune each detection method with hyperparameters optimized based on their respective papers. The detailed implementations for all competing defenses are given below:

**NC** [45]. For the implementation of NC (Neural Cleanse), we follow the official code released by Wang et al. (2019). The method searches for minimal perturbation patterns that cause misclassification to a specific target class, and flags potential backdoor behavior if the required perturbation is significantly smaller than others. We apply NC to detect backdoor triggers after the victim model is trained.

**ABS** [35]. We adopt the official implementation of ABS (Activation Clustering-Based Signature), which identifies potential backdoored neurons by analyzing the neuron activation distribution across clean and poisoned samples. A strong activation pattern discrepancy indicates the presence of a backdoor. We use TinyImageNet as the evaluation dataset and apply ABS on the final convolutional layer.

**NAD** [26]. For NAD (Neural Attention Distillation), we follow the setup in the original paper. NAD defends against backdoors by distilling knowledge from a suspicious model into a student model using attention transfer, which helps suppress backdoor behaviors. We use the public NAD codebase and apply it after finetuning with a small clean subset.

**UNICORN** [48]. For the implementation of UNICORN, we follow the official code. UNICORN is a unified backdoor trigger inversion framework designed to recover potential backdoor triggers from a trained victim model without access to the original training data. It optimizes a trigger pattern and mask jointly by minimizing classification loss on a clean validation set while maximizing the attack success rate on target labels. We apply UNICORN on the TinyImageNet dataset using a ResNet-18 backbone, initializing trigger size and mask as suggested in the original paper, and report the recovered trigger quality and subsequent defense efficacy.

**MM-BD** [46]. For the implementation of MM-BD (Maximum Margin Backdoor Detection), we follow the official code and experimental protocol. MM-BD is a post-training backdoor detection method designed to identify backdoored models regardless of the trigger pattern type by leveraging a maximum margin statistic computed on the penultimate layer features. The method effectively distinguishes clean and backdoored classes by analyzing class-wise feature margins. Due to its strong transferability, we extend MM-BD to the vision-language model (VLM) setting and evaluate its detection performance on COCO datasets.

**DECREE** [12]. For the implementation of DECREE, we follow the official code and methodology. DECREE is designed to detect backdoors in pre-trained encoders by analyzing the encoder's latent representations and identifying anomalous patterns associated with backdoor triggers. The method

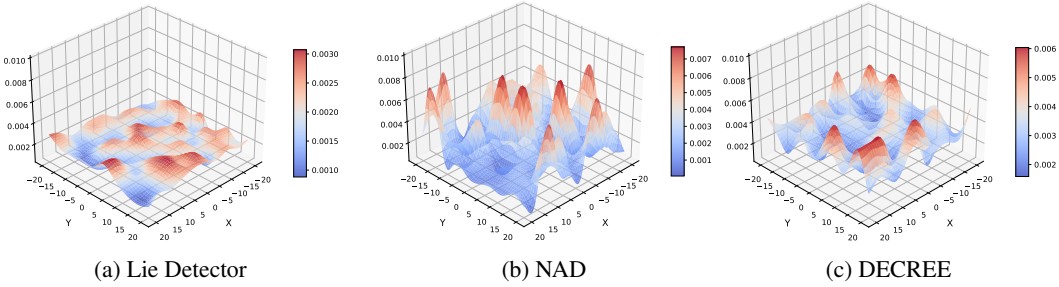

(a) Lie Detector        (b) NAD        (c) DECREE

Figure Appendix 1: Stability of different defense methods on Blended.

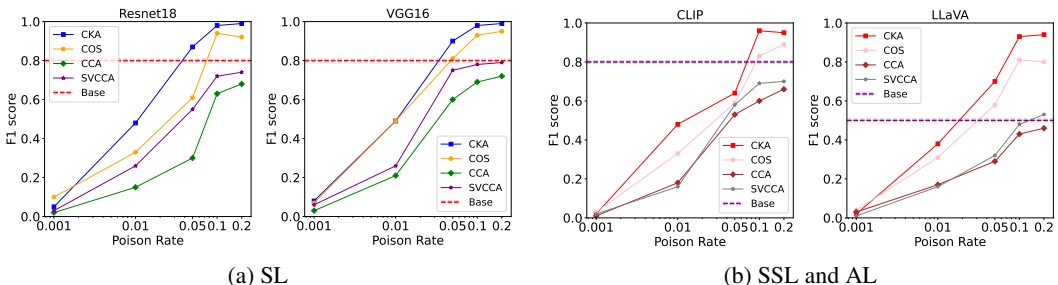

(a) SL                  (b) SSL and AL

Figure Appendix 2: F1 scores under four different similarity metrics.

does not require access to the original training data and can be applied post-hoc on the encoder. We evaluate DECREE on the Caltech101 dataset with a CLIP (backbone: ResNet-50), reporting detection accuracy and robustness across multiple backdoor attack variants.

**SEER** [55]. For the implementation of SEER, we follow the official code and experimental setup. SEER is a backdoor detection framework tailored for vision-language models, which jointly searches for target text triggers and corresponding image triggers to identify backdoor behaviors. The method exploits multimodal correlations to effectively detect poisoned inputs without requiring prior knowledge of the trigger patterns. We evaluate SEER on a variety of datasets, reporting detection accuracy and false positive rates under various backdoor attacks.

## C  Method stability

We conducted 10 experiments to obtain F1 scores, from which a variance was calculated. A total of 100 tests were performed, resulting in 10 sets of variances, which were used to evaluate the stability of the method, as shown in Figure Appendix 1.

## D  Effect of Poison Rate and Similarity Metric on Detection Performance

To analyze how poison rate and similarity metrics affect trigger reverse, we report the F1 scores of four similarity metrics (CKA, COS, CCA, SVCCA) across four model architectures in Fig. Appendix 2. The evaluation covers supervised models (ResNet-18, VGG16), contrastive models (CLIP), and multimodal models (LLaVA), tested under five poison rates (0.1%, 1%, 5%, 10%, 20%) against the Blended attack. Higher F1 scores indicate better detection performance.

We highlight three key observations: (1) **CKA achieves the highest F1 scores across all settings**, significantly outperforming COS, CCA, and SVCCA. This demonstrates CKA's robustness in capturing backdoor-induced representation shifts across different architectures and learning paradigms. (2) **Detection performance improves with higher poison rates**. All metrics show increasing F1 scores as the poison rate rises. However, traditional metrics struggle in low-poison regimes, while CKA maintains strong performance even at 0.1% and 1%, validating its sensitivity to subtle backdoor effects. (3) **At extremely low poison rates**, detection becomes difficult due to the weak backdoor effect and limited number of poisoned samples. In these cases, the ASR remains below 10–20%,

Table Appendix 3: Levels of model collusion simulated in our experiments.

| Level | Name | Description | Data Share | Initialize | LR | Epoch |
|---|---|---|---|---|---|---|
| L0 | Baseline | Same backdoor target and config, fully independent training | ✗ | ✗ | ✓ | ✓ |
| L1 | Weak | Partial backdoor data share (30% overlap); different initialization | 30% | ✗ | ✓ | ✓ |
| L2 | Moderate | Full backdoor data share; different initialization | ✓ | ✗ | ✓ | ✓ |
| L3 | Strong | Full backdoor data share and identical training | ✓ | ✓ | ✓ | ✓ |

Table Appendix 4: Lie Detector performance under varying levels of model collusion.

| Attack / Dataset | Model | Level | DSR (%) | FPR (%) | SOTA DSR / FPR (%) |
|---|---|---|---|---|---|
| ISSBA / CIFAR-10 (MM-BD) | ResNet-18 | L0 | 100.0 | 0.0 | 92.5 / 5.0 |
| | | L1 | 95.0 | 2.5 | |
| | | L2 | 82.5 | 7.5 | |
| | | L3 | 70.0 | 15.0 | |
| BadCLIP / Caltech101 (SEER) | CLIP | L0 | 95.0 | 5.0 | 90.0 / 5.0 |
| | | L1 | 90.0 | 5.0 | |
| | | L2 | 75.0 | 10.0 | |
| | | L3 | 72.5 | 15.0 | |
| TrojanVLM / COCO (SEER) | LLaVA | L0 | 92.5 | 5.0 | 80.0 / 15.0 |
| | | L1 | 85.0 | 7.5 | |
| | | L2 | 72.5 | 12.5 | |
| | | L3 | 67.5 | 17.5 | |

and the backdoored model behaves similarly to a clean model, leading to low F1 scores across all methods. Nonetheless, such low poisoning also implies minimal real-world threat, as the attack itself is largely ineffective.

# E  Advanced scenarios

We discuss three advanced scenarios here to assess the applicability and limitations of *Lie Detector* beyond the untrusted third party setting:

**1)** *Collusion Attacks.* To further evaluate the robustness of **Lie Detector** under collusion attacks, we simulate different levels of *provider collusion*. As summarized below, "✓"/✗ indicate whether two providers share or differ in a given training aspect:

Our method remains robust under **weak or moderate collusion (L1–L2)**. Even when some poisoned data are shared between providers, the models still learn distinct internal representations, and **CKA** successfully captures these discrepancies. Consequently, **Lie Detector** continues to outperform SOTA detection approaches in these settings.

As expected in Tab. Appendix 4, performance degrades under **strong collusion (L3)**, where both models are nearly identical due to shared initialization and training data. This extreme setting, discussed in Appendix 3, represents an impractical yet instructive worst-case scenario. Importantly, even under such strong collusion, **Lie Detector** maintains a **detection success rate exceeding 70%**, demonstrating resilience against synchronized adversaries.

In real-world deployments, such perfectly coordinated collusion is rare and difficult to achieve, especially among independent commercial or decentralized service providers. One promising mitigation strategy is to **randomly distribute training data among multiple independent parties**, reducing the risk of collusion from the outset. Later, their models can be aggregated via **ensemble or knowledge distillation**, maintaining accuracy while enhancing robustness. We plan to explore this direction in future work.

**2)** *Client-Level Inspection in Federated Learning (FL).* We consider a realistic FL-inspired use case, where models from multiple clients are submitted for centralized auditing before aggregation. Using CIFAR-10, we divide the training set into four equal partitions and assign them to four independent clients. Some clients train clean ResNet-18 models, while the remaining clients apply backdoor

Table Appendix 5: Detection performance of Lie Detector under FL-inspired client-level inspection on CIFAR-10. Each cell shows DSR / FPR (%). The header denotes the number of backdoored clients out of 4 total clients.

| Attack ↓, $\frac{\text{backdoored\_clients}}{\text{client\_num}}$ → | 0/4 | 1/4 | 2/4 | 3/4 | 4/4 |
|---|---|---|---|---|---|
| BadNet | 100.00 / 0.00 | 99.17 / 0.00 | 98.50 / 0.00 | 97.33 / 0.00 | 96.17 / 0.33 |
| Blended | 100.00 / 0.00 | 98.00 / 0.00 | 98.17 / 0.33 | 97.67 / 0.50 | 96.00 / 0.83 |
| ISSBA | 100.00 / 0.00 | 96.17 / 0.83 | 96.67 / 0.50 | 97.00 / 0.67 | 95.83 / 1.33 |
| Average | 100.00 / 0.00 | 97.78 / 0.28 | 97.78 / 0.28 | 97.33 / 0.39 | 96.00 / 0.69 |

attacks (BadNet, Blended, ISSBA, and Low-Frequency). Each client trains its model locally without any parameter sharing or global model fusion. We then evaluate Lie Detector by exhaustively sampling model pairs (4×3=12 combinations) and performing detection over 50 trials.

Tab. Appendix 5 shows the detection success rate (DSR) and false positive rate (FPR) under varying numbers of backdoored clients, denoted as `backdoored_client / client_num`. We simulate scenarios from fully clean (0/4) to fully poisoned (4/4), offering a comprehensive view under different FL threat levels. Lie Detector remains robust across all settings. In the clean case (0/4), it correctly raises no alarms (FPR = 0%, DSR = 100%). As backdoored clients increase (1/4 to 3/4), DSR stays high (96.17%–99.17%) and FPR remains low (≤1%), indicating strong sensitivity to injected backdoors without misclassifying clean models. Even in the hardest case (4/4), where no clean client exists, the method still achieves >95% DSR and <1.5% FPR across all attack types. This suggests Lie Detector can exploit subtle inconsistencies from imperfect backdoor optimization, even among colluding clients. Slightly higher FPRs are observed for ISSBA and Low-Frequency in high-poisoning scenarios, reflecting their stealthy nature, but overall resilience holds. These results demonstrate Lie Detector's effectiveness for decentralized auditing in FL without requiring clean references, aggregation, or inter-client communication.

**3) *Scaling to Larger Models.*** We further assess Lie Detector on high-capacity vision-language models: VisualGLM-6B [9] and LLaVA-1.5-7B [24], whose GFLOPs are 191.1 and 76.6, respectively. We adopt two recent multi-modal backdoor attacks TrojanVLM and Shadowcast, and construct 20 clean and 20 poisoned models per model-attack combination via fine-tuning with or without injected triggers. The experimental setup is same as the main paper. The results are in Tab. Appendix 6. On VisualGLM-6B, Lie Detector achieves a DSR of 90.0% and FPR of 5.0% under TrojanVLM, and 85.0% DSR and 10.0% FPR under Shadowcast. These results confirm that Lie Detector generalizes well to larger high-capacity backdoored models, making it a promising solution for securing next-generation foundation architectures.

Table Appendix 6: Detection results on large-scale VLMs under TrojanVLM and Shadowcast attacks. We use 20 clean and 20 poisoned models for evaluating VisualGLM-6B and LLaVA-1.5-7B.

| Model | GFLOPs | Attack | DSR (%) | FPR (%) |
|---|---|---|---|---|
| VisualGLM-6B | 191.1 | TrojanVLM | 90.00 | 5.00 |
| | | Shadowcast | 85.00 | 10.00 |
| LLaVA-1.5-7B | 76.6 | TrojanVLM | 92.50 | 0.00 |
| | | Shadowcast | 90.00 | 5.00 |

# F   Detailed Comparisons with Existing Backdoor Detection Methods

To provide a comprehensive understanding of the strengths of our method, we compare **Lie Detector** with several representative backdoor detection techniques, including post-training methods (MM-BD, NAD, ABS, NC, UNICORN) and the pre-training method DECRE. The comparison covers multiple aspects such as computational cost, label and data dependency, applicable scenarios, limitations, and detection performance. A detailed summary is presented in Tab. Appendix 7.

Table Appendix 7: Comparison with existing backdoor detection methods. Cost: Computational Cost. Label: whether ground-truth labels are required. Performance: average DSR across datasets (from Tab. 1).

| Method | Cost | Label Required | Data Dependency | Applicable Scenario | Limitation | Performance |
|---|---|---|---|---|---|---|
| MM-BD | Low | No | No clean data | Post-training | Weak on adaptive attacks | 94.7% |
| NAD | High | Yes | Clean data needed | Mitigation | High cost | 53.1% |
| ABS | High | Yes | Clean data needed | Detection | Poor for spatial triggers | 52.5% |
| NC | Medium | Yes | Clean data needed | Detection | Poor for global triggers | 32.5% |
| UNICORN | High | No | Clean data needed | Multi-trigger detection | High cost | 81.6% |
| DECREE | Low | No | No clean data | Pre-training (SSL/multimodal) | Pre-training only | 92.8% |
| **Lie Detector** | Medium | No | Clean data only | Unified (SL/SSL/AL) | Assumes two models | **99.7%** |

As shown in the table, many existing methods rely on ground-truth labels and clean training data, which may not always be available in practical scenarios. Several also operate under the white-box assumption or require training dynamics, making them less applicable to black-box or third-party verification settings.

In contrast, **Lie Detector** does not require label supervision or access to model internals, and is applicable across supervised, self-supervised, and autoregressive learning paradigms. It achieves state-of-the-art performance (99.7% DSR) while maintaining moderate computational cost, and uniquely supports unified detection in complex settings like multimodal LLMs. Also, as deonstrated in the main paper, our method has extremely low false positive rate.

The only minor limitation is the requirement of two independently trained models, which is a reasonable and realistic assumption third-party scenarios.

# G Theoretical Properties of Similarity Metrics

## G.1 Summary of Properties

We compare four commonly used similarity metrics Cosine similarity, Canonical Correlation Analysis (CCA), Singular Vector CCA (SVCCA), and Centered Kernel Alignment (CKA) across key theoretical properties. The comparison is summarized in Tab. Appendix 8.

Table Appendix 8: Comparison of theoretical properties across similarity metrics.

| Metric | Scale Invariant | Angle Sensitive | Cross-Model Stable | Nonlinear Compatible |
|---|---|---|---|---|
| Cosine | No | Yes | Low (basis sensitive) | No |
| CCA | No | No | Medium (linear only) | No |
| SVCCA | Partial | No | Medium (SVD improves stability) | No |
| CKA | Yes | Yes | High | Yes |

Among all the evaluated similarity metrics, CKA uniquely satisfies all four desirable theoretical properties: it is invariant to isotropic scaling, sensitive to angular alignment, robust to architectural changes, and compatible with nonlinear relationships. These strengths are especially critical in our setting, where models may differ in architecture, training dynamics, or feature scales. In contrast, Cosine similarity lacks stability across bases, CCA fails under nonlinearity, and SVCCA only partially improves robustness through dimensionality reduction. CKA's kernel-based formulation and normalization by Frobenius norm ensure consistent and meaningful comparisons across diverse model outputs, making it particularly well-suited for cross-model backdoor detection in the absence of clean references. We also provides the mathematical proofs in the following section.

## G.2 Mathematical Proofs

### 1. Cosine Similarity [43, 36]

**Definition:** Given vectors $\mathbf{a}, \mathbf{b} \in \mathbb{R}^d$, cosine similarity is defined as:

$$\text{Cos}(\mathbf{a}, \mathbf{b}) = \frac{\langle \mathbf{a}, \mathbf{b} \rangle}{\|\mathbf{a}\| \cdot \|\mathbf{b}\|}$$

**Property Proofs**

- **Scale Invariance:** Cosine similarity is invariant to positive scalar multiplication:

$$\text{Cos}(\lambda \mathbf{a}, \mathbf{b}) = \frac{\lambda \langle \mathbf{a}, \mathbf{b} \rangle}{\lambda \|\mathbf{a}\| \cdot \|\mathbf{b}\|} = \text{Cos}(\mathbf{a}, \mathbf{b})$$

  However, this does not hold under general affine or non-uniform scaling. It also fails under feature shuffling or reparametrization.

- **Angle Sensitivity:** Cosine similarity explicitly measures $\cos(\theta)$, the angle between $\mathbf{a}$ and $\mathbf{b}$. For unit vectors:

$$\text{Cos}(\mathbf{a}, \mathbf{b}) = \cos(\theta)$$

- **Cross-Model Stability:** Cosine similarity is sensitive to feature basis. A rotation matrix $R$ gives:

$$\text{Cos}(R\mathbf{a}, \mathbf{b}) \neq \text{Cos}(\mathbf{a}, \mathbf{b})$$

- **Nonlinear Compatibility:** Not compatible. Cosine similarity is a linear measure and does not preserve structure under nonlinear transformations.

## 2. Canonical Correlation Analysis (CCA) [17, 14]

**Definition:** Given two centered data matrices $X \in \mathbb{R}^{n \times p}$ and $Y \in \mathbb{R}^{n \times q}$, CCA finds projections $w_x \in \mathbb{R}^p$, $w_y \in \mathbb{R}^q$ that maximize the correlation between $Xw_x$ and $Yw_y$:

$$\rho = \max_{w_x, w_y} \frac{w_x^\top \Sigma_{XY} w_y}{\sqrt{w_x^\top \Sigma_{XX} w_x} \cdot \sqrt{w_y^\top \Sigma_{YY} w_y}}$$

**Property Proofs**

- **Scale Invariance:** If $X' = DX$ for diagonal $D$, then:

$$\Sigma_{X'X'} = D\Sigma_{XX}D^\top, \quad \Sigma_{X'Y} = D\Sigma_{XY}$$

  The correlation changes unless $D = \lambda I$, i.e., only isotropic scaling is invariant. Hence CCA is not generally scale-invariant.

- **Angle Sensitivity:** CCA finds directions maximizing correlation, not angle:

$$\text{corr}(Xw_x, Yw_y) \neq \cos(\theta)$$

  No explicit relation to angular alignment $\rightarrow$ not angle-sensitive.

- **Cross-Model Stability:** Sensitive to changes in basis; aligned projections across independently trained models are not guaranteed unless architectures match.

- **Nonlinear Compatibility:** CCA is linear; incapable of capturing nonlinear dependencies.

## 3. SVCCA [41]

**Definition:** SVCCA applies singular value decomposition to reduce noise, then uses CCA. Let $X \in \mathbb{R}^{n \times p}$:

$$X = U_X S_X V_X^\top, \quad \text{keep top } k \text{ components}$$

Apply CCA on $U_X^k$, $U_Y^k$.

**Property Proofs**

- **Scale Invariance:** If $X \rightarrow \lambda X$, then $S_X \rightarrow \lambda S_X$ and $U_X$ is unchanged. So SVD step is scale-invariant. But since CCA is not, SVCCA is only partially scale-invariant.

- **Angle Sensitivity:** CCA is used after SVD. Since neither SVD nor CCA is angle-sensitive, SVCCA is not.

- **Cross-Model Stability:** SVD suppresses noise and basis sensitivity. Better than CCA.

- **Nonlinear Compatibility:** Still linear; no support for nonlinearity.

## 4. Centered Kernel Alignment (CKA) [5, 20, 1]

**Definition:** Given two activation matrices $A_1, A_2 \in \mathbb{R}^{n \times p}$ (rows are samples), define their Gram (kernel) matrices:

$$K_1 = H A_1 A_1^\top H, \quad K_2 = H A_2 A_2^\top H,$$

where $H = I - \frac{1}{n} \mathbf{1} \mathbf{1}^\top$ is the centering matrix that removes the mean from each feature vector.

Then the linear CKA similarity is defined as:

$$\text{CKA}(A_1, A_2) = \frac{\langle K_1, K_2 \rangle_F}{\|K_1\|_F \cdot \|K_2\|_F},$$

where $\langle A, B \rangle_F = \text{Tr}(A^\top B)$ is the Frobenius inner product and $\|A\|_F = \sqrt{\langle A, A \rangle_F}$ is the Frobenius norm.

**Property Proofs:**

- **Scale Invariance:** Suppose $A_1 \mapsto \lambda A_1$ and $A_2 \mapsto \mu A_2$, with scalars $\lambda, \mu \in \mathbb{R}$. Then:

$$K_1 \mapsto \lambda^2 H A_1 A_1^\top H = \lambda^2 K_1, \quad K_2 \mapsto \mu^2 K_2$$

  Hence:

$$\text{CKA}(\lambda A_1, \mu A_2) = \frac{\lambda^2 \mu^2 \langle K_1, K_2 \rangle_F}{\lambda^2 \|K_1\|_F \cdot \mu^2 \|K_2\|_F} = \text{CKA}(A_1, A_2)$$

  Therefore, CKA is invariant to isotropic scaling of inputs.

- **Orthogonal Invariance:** Suppose $A_1 \mapsto A_1 Q$ and $A_2 \mapsto A_2 R$ where $Q, R$ are orthogonal matrices (i.e., $Q^\top Q = I$, $R^\top R = I$). Then:

$$A_1 A_1^\top \mapsto (A_1 Q)(A_1 Q)^\top = A_1 Q Q^\top A_1^\top = A_1 A_1^\top$$

  Hence, $K_1$ and $K_2$ remain unchanged $\rightarrow$ CKA is invariant to orthogonal transformations (rotations, reflections, etc.).

- **Angle Sensitivity:** Since the Frobenius inner product between two kernel matrices $K_1$ and $K_2$ reflects alignment between their feature spaces:

$$\langle K_1, K_2 \rangle_F = \sum_{i,j=1}^{n} K_1(i,j) \cdot K_2(i,j),$$

  it is maximized when the two representations encode similar pairwise distances (i.e., angles) between samples.

  Moreover, when the features are centered and normalized, CKA behaves similarly to cosine similarity in the kernel (pairwise similarity) space:

$$\text{CKA} = \cos \angle (K_1, K_2),$$

  making it sensitive to representational misalignment.

- **Cross-Model Stability:** Due to centering (which removes mean differences) and Frobenius normalization (which removes magnitude differences), CKA is robust across model architectures, feature dimensionalities, and training dynamics.

  It is also **basis-invariant**, meaning it evaluates the relative structure between representations regardless of coordinate systems:

$$\text{CKA}(A, A) = 1, \quad \text{CKA}(A, B) < 1 \text{ iff representations differ.}$$

- **Nonlinear Compatibility:** CKA is compatible with nonlinear feature mappings. For example, let $\phi : \mathbb{R}^p \to \mathcal{H}$ be a nonlinear map to a high-dimensional (possibly infinite) Hilbert space. Then kernel matrices are computed via:

$$K_{ij} = \langle \phi(x_i), \phi(x_j) \rangle$$

  allowing CKA to measure similarity in both linear and nonlinear spaces by replacing $A_i A_i^\top$ with any positive semidefinite kernel $K_i$.

# H Impact Statement

This work proposes a practical and general-purpose framework for detecting backdoors in machine learning models, particularly in outsourced or third-party training settings. The proposed cross-examination mechanism eliminates the need for a trusted clean model or prior attack knowledge, enabling robust verification across supervised, self-supervised, and autoregressive learning paradigms. Notably, it is the first to support backdoor detection in large multimodal vision-language models (*e.g.,*LLaVA, MiniGPT-4), addressing a critical gap in securing foundation models. From a broader perspective, this work contributes to the growing need for AI accountability and trustworthy deployment, especially as AI models are increasingly developed by external vendors or deployed in critical applications such as healthcare, finance, and national security. By reducing reliance on assumptions about attackers or training access, the framework enhances the resilience of model verification protocols. On the social level, this research promotes transparency and auditability in machine learning pipelines, aligning with global efforts around AI governance and certification. While the method can expose malicious behaviors, it does not introduce harm, manipulate data, or compromise privacy. Nevertheless, continued evaluation is needed to ensure fairness in model comparisons and avoid mislabeling benign discrepancies as malicious behavior in edge cases.

# I Limitation

Our framework assumes an untrusted third-party verification setting, where third-party providers independently train models and do not actively collude. While this assumption holds in many real-world applications, such as government or enterprise auditing, AutoML pipelines, or federated deployments with disjoint training, it may not capture stronger threat models. For instance, in collusion attacks, coordinated adversaries may align backdoored models to mask inconsistencies, potentially reducing the effectiveness of cross-model comparison. We note that in the extreme case where two backdoored models fully collude, the detection success rate (DSR) drops significantly. For example, under the BadNet/CLIP setting, we observe that the DSR can decrease sharply to 45%. This indicates that collusive backdoors constitute a current limitation of our method, warranting further investigation and the development of more robust attack and defense schemes in future work.

Furthermore, in this work, all attack and defense evaluations on multimodal models follow experimental settings consistent with prior studies, primarily performing poisoning or fine-tuning at the project layer or downstream head, while keeping the CLIP and LLM backbones frozen during the poisoning stage to ensure methodological comparability and reproducibility. We explicitly report, for each experiment, the targeted layer and the range of trainable parameters (including LoRA, Adapter, downstream fine-tuning, and full fine-tuning strategies).

It is important to emphasize that if full-parameter fine-tuning or alternative fine-tuning hierarchies are adopted, it becomes necessary to separately measure and jointly analyze the impact on key CLIP and LLM modules, as well as the corresponding attack and defense performance. Since this study does not provide a systematic evaluation covering all modules under full fine-tuning, this constitutes a limitation of our current work. Future research will extend this analysis with more comprehensive module-level comparisons and evaluations to achieve a deeper understanding of model robustness across different fine-tuning paradigms.

These scenarios point to promising directions for future work rather than fundamental limitations, and our framework offers a solid foundation for extending to such advanced settings.

