# OpenReview forum: "Lie Detector: Unified Backdoor Detection via Cross-Examination Framework"
_NeurIPS.cc/2025/Conference — NeurIPS 2025 poster_

### Official Review · Reviewer_3ggA · 2025-06-25

**Clarity:** 2
**Significance:** 2
**Originality:** 2
**Rating:** 4
**Confidence:** 4

**Summary:**

This paper proposes a backdoor detection approach in a `semi-honest’ out-sourcing training scenario. Users trust a third party to train models by providing their data, model and protocols. After receiving trained models from different out-sourcing parties, the proposed detection method aims to find the potential backdoored model between the received ones. The method optimizes over backdoor triggers from independent training results from two service providers, and exploits criteria including CKA for model similarity and distributions to perform the backdoor detection. It is generalizable in supervised and semi-supervised learning. Essentially, this method is built upon two training losses: loss of output distribution, which is constructed based on the label or clean reference; and loss of CKA, which is measured as a similarity score for a specific intermediate layer between independent models. After required the trigger, the proposed method distinguishes the backdoor model by testing it on a poisoned dataset and a clean dataset, by measuring the ASR against a given threshold.

**Questions:**

1.	Can authors clarify the attacker/defender knowledge/capabilities referring to the cited previous works?
2.	Can authors clarify the result/consequence if the adversary is assumed to have the knowledge of detection method.
3.	What if two service providers collude and injects the same backdoor?
4.	Please clarify because I'm confused by the claim that the detection requires no access to its training data, but algorithm says otherwise.
5.	Please clarify if anything is incorrect in the weaknesses part.

**Ethical Concerns:**

["NO or VERY MINOR ethics concerns only"]

**Final Justification:**

Most of my concerns are addressed, while some strong assumptions require further evaluation and systematic study.

**Limitations:**

Yes.

**Paper Formatting Concerns:**

None.

**Quality:**

2

**Strengths And Weaknesses:**

Strengths:

1.	The motivation of the proposed detection mechanism is clearly stated and indeed needs further research.

2.	The proposed method is applicable to different learning paradigms including supervised learning, semi-supervised learning, and autoregressive learning.

3.	The proposed method is able to defend against imperceptible or frequency based backdoor triggers, which outperforms previous defense mechanisms.

Weaknesses:

1.	Unclear threat model: The proposed adversary is NOT a semi-honest adversary. Usually, semi-honest refers to a curious-but-honest setting, though authors refer to adversary controls the backdoor training but not the detection. This is a malicious setting. Here, the author needs to check previous works and clarify the threat model in a standard way.

2.	Strong assumptions:
a)	In this work, adversary is assumed to be unaware of the detection method. Authors may refer the threat model assumption in two works, TED, MM-BD, from S&P, which are both cited in this paper. They both assume the attacker has full knowledge of the defense mechanism, which is commonly assumed for a defense work. Especially, authors should NOT claim that the proposed method is robust to adaptive attacks by assuming the unawareness of detection. Please refer to existing defense works.
b)	What if two service providers collude and injects the same backdoor? In this paper, one of the service providers is malicious, who deviates from the given protocol and injects the backdoor, while the other service provider is honest and follow the protocol. But if the same or similar backdoors are implemented to two models, what will happen? This case should have been encountered in the experiments based on configuration provided in the paper, but it seems no ablation study about this issue is provided.

3.	Dependency on clean dataset. The proposed detection method claims to require no access to its training data. However, the training set D_s is sampled as clean instances from D_c.

4.	Experiment settings unfairness. In the configuration, given a pool of N models, authors sample every single pair from the model pool, and perform the detection on each pair, then uses a majority vote to determine whether a model is backdoored. First, the threshold of majority vote is not given (poison rate of the pool should be unknown). Second, I reckon such evaluation criteria for the proposed detection might have deviated from the original setting that two service providers independently train two models with one of them injected with a backdoor. Because in the original setting, there are no N models to compare with. Such evaluation based on majority vote could give extra advantage to the proposed detection, which undermines the comparison fairness against other baselines.

5.	Generalizability. The detection method is generalized to supervised learning, semi-SL and autoregressive learning. The OD loss always requires a clean reference to construct it. Without such reference, like the GT label, this method does not work or might fail in self-supervised learning that is without any clean reference.

---

> ### Author Rebuttal · Authors · 2025-07-31
>
> **W1&Q1**. Unclear threat model: The paper claims a semi-honest adversary, but the adversary controls training and may embed backdoors — this is not honest-but-curious.
>
> **A1**. Thanks. We misused the term "semi-honest adversary": while the notion of a potentially manipulated training process is widely adopted in the backdoor detection literature, the term "semi-honest" has a strict definition in cryptography (i.e., honest but curious), which we overlooked.
>
> In fact, the threat model we adopt is more accurately described by the machine learning community’s notion of an “untrusted third-party provider”: the model is trained externally by a service provider who may deviate from the intended protocol and insert backdoors, while detection is performed independently by the user (a trusted entity). The attacker (untrusted provider) has full control over training but no knowledge of the detection method, while the defender (user) has no access to training data or process—consistent with prior works such as DECREE.
>
> Accordingly, we will revise the manuscript as follows: Replace all mentions of "semi-honest adversary" with "untrusted third-party provider" to avoid ambiguity.
>
> Modify the beginning of Section 3.1 (“Threat Model”) to:
>
> *We consider a practical outsourced training scenario, where model training is performed by a third-party provider who may nominally follow the protocol but is essentially untrusted and may insert backdoors. The user has no visibility into the training process, but can independently inspect the returned model for malicious behavior.*
>
> **W2&Q2&Q3**. Strong assumptions. (a) The paper assumes a non-adaptive attacker unaware of the detection, unlike prior works with adaptive settings, weakening robustness claims. (b) Collusion between providers injecting identical backdoors is realistic but not evaluated.
>
> **A2 (a)**. We follow the adaptive attacks similar to TED and MM-BD, which simulates adaptive attackers by modifying the training process of the backdoor model.
>
> We design an adaptive attacker that is aware of our detection and aims to bypass our CKA-based representation divergence detector. In particular, the attacker first trains a clean reference model $f_{clean}$ on the clean training dataset. Then, when training the adaptive backdoored model $f$ on the poisoned training dataset, the attacker combines the original training loss $L_{origin}$ with a regularization term designed to suppress detection. This regularizer maximizes the CKA similarity between the clean and backdoored models:
>
> $$
> L_{adaptive} =L_{origin} - \lambda \cdot L_{CKA}(f, f_{clean})
> $$
>
> $L_{CKA}$ denotes the CKA similarity between intermediate representations of the two models under the same input as used in our paper, and $\lambda$ controls the strength. We vary the $\lambda$ and report the attacker’s ASR as well as our defense’s DSR and FPR in the table below.
> |λ|ASR(%)|DSR(%)|FPR(%)|
> |-|-|-|-|
> |0 (No adaptive)|98.75|99.61|0.26|
> |0.1|94.90|91.54|1.79|
> |0.2|90.68|90.26|3.72|
> |0.5|82.37|86.54|9.10|
>
> From the results, we conclude:
> - **Adaptive attacks do reduce detection accuracy:** As $\lambda$ increases, the attacker can partially suppress cross-model representation divergence, causing our DSR to degrade from 99.61% to 86.54%.
> - **The attacker must sacrifice backdoor effectiveness:** The ASR drops from 98.75% to 82.37%, indicating that the backdoor behavior itself is weakened. This reveals a trade-off between stealth and efficacy，as attackers cannot simultaneously achieve high attack success and high stealth.
>
> In summary, our method remains robust even under defense-aware, adaptive threats, raising the barrier and cost for stealthy backdoor injection.
>
> **A2 (b)**. We believe there may be a misunderstanding. In our experiments, we randomly sample model pairs from a pool containing both clean and backdoored models. Thus, all three combinations are covered: **clean-clean**, **backdoor-backdoor**, or **clean-backdoor**. Importantly, backdoored models share the same injected trigger. The strong performance reported in our paper reflects our method's robustness under all these possible pairings.
> 1) In **Appendix E**, we consider the **strongest model collusion scenario**, where **two service providers share the same poisoned data, model initialization, and training strategy** to implant **identical backdoors**. In our response **A1** to **Reviewer VC1P**, we proposed a more **systematic evaluation for collusion-based backdoor attacks**, introducing four levels ranging from **L0 (completely independent)** to **L3 (strong collusion)**. We conducted experiments on three attacks and three tasks. Results show that our method maintains performance superior to existing SOTA methods even under weak collusion (L1), such as achieving an 85% DSR on TrojanVLM compared to the existing method's 80%.
> 2) Collusion is permitted in extreme cases, to reduce the risk of detection failure, we randomly assign training data to independent parties, ensuring their backdoor samples do not overlap from the source.
>
>
>
> **W3&Q4**. Clean data dependency unclear: $\mathcal{D}_s$ is sampled from $\mathcal{D}_c$, which may imply access to clean data.
>
> **A3**. In the original manuscript, we stated:
> *“The verifier user (the police) has no access to the training data or process and cannot assume the availability of a clean reference model.”*
> We acknowledge that this may cause confusion. To clarify: in our setting, the verifier **cannot access the full training dataset used by the providers**, nor can they observe the actual training process. However, the verifier may possess a **small subset of clean data**, either from their own held-out inputs or collected from the same domain. This subset is used for trigger optimization and detection.
> We revise the statement for clarity as:
> > *“The verifier cannot access the full training data or training process on the provider side but may have access to a small, trusted subset of clean data. No clean reference model is assumed.”*
>
> This accurately reflects our assumption: detection relies on a small amount of **verifier-held clean data**, not on any privileged access to the full training set.
>
>
>
> **W4**. Evaluation unfair: majority vote assumes multiple models, deviating from the original two-provider setting.
>
> **A4**. Thanks. Below we clarify the intent and fairness of our evaluation design:
>
> In “Configuration”, “We randomly select $N(N-1)/2$ pairs of models from $N$ models and apply voting.” What we actually meant was: From the $C(N, 2)$ possible pairs of models formed by the $N$ models, we randomly select 20 model pairs (without repetition) for testing, and compute the detection performance by averaging scores. We use “voting” in the original text was inaccurate. In the initial draft, the term “voting” was primarily used to summarize the concept of “enhancing robustness through multiple comparisons,” but it failed to accurately distinguish between soft aggregation and hard majority voting. Thanks for pointing out this error and have unified the terminology in the final version to “average of scores from multiple comparisons.”
> Additionally, to further address the concern about the fairness of the evaluation, we include results using all $C(40, 2) = 780$ model pairs, and then average the results.
>
> # ResNet-18
>
> |Dataset|Attack|DSR|FPR|SOTA DSR|SOTA FPR|
> |-|-|-|-|-|-|
> |CIFAR-10|BadNet|100.00|0.00|100.0|0.0|
> ||Blended|100.0|0.00|97.5|0.0|
> ||ISSBA|99.61|0.26|95.0|10.0|
> ||Low-Frequency|99.23|1.03|92.5|5.0|
> |TinyImageNet|BadNet|99.36|0.0|97.5|0.0|
> ||Blended|98.84|0.90|95.0|0.0|
> ||ISSBA|97.56|1.54|95.0|5.0|
> ||Low-Frequency|98.21|1.92|92.5|5.0|
>
> # CLIP
>
> |Dataset|Attack|DSR|FPR|SOTA DSR|SOTA FPR|
> |-|-|-|-|-|-|
> |Caltech101|BadNet|98.08|0.51|95.0|5.0|
> ||Blended|97.31|2.56|97.5|5.0|
> ||BadCLIP|94.61|4.62|90.0|0.0|
> |ImageNet|BadNet|96.67|0.25|95.0|0.0|
> ||Blended|93.46|0.38|90.0|5.0|
> ||BadCLIP|90.13|4.23|87.5|10.0|
>
> # LLaVA
>
> |Dataset|Attack|DSR|FPR|SOTA DSR|SOTA FPR|
> |-|-|-|-|-|-|
> |COCO|TrojanVLM|94.48|5.13|80.0|15.0|
> ||ShadowCast|92.18|0.51|85.0|10.0|
> |Flickr30K|TrojanVLM|90.51|7.05|80.0|5.0|
> ||ShadowCast|90.26|4.74|80.0|10.0|
>
>
> Compared with the results in the main text, the average results did not show a significant decline in performance. The detection success rate and false positive rate remained stable, with an average variance of less than 0.5, and **consistently outperformed all baseline methods* across all tasks and model types.
>
> **W5**. Generalizability — OD Loss seems to require clean labels, which may not be available in self-supervised settings.
>
> **A5**. We clarify that the OD Loss in our framework is **not tied to class labels or external references**, but is instead defined as the **negative of the original task loss** under each learning paradigm (Eq. (5)).
>
> In backdoor attacks against **self-supervised learning (SSL)** scenarios such as **MoCo v2**, **BYOL**, or **MSF**, where no labels are available [1], we use the **negative of the SSL training loss** (e.g., contrastive objectives). These losses are inherently label-free, making OD Loss applicable in such settings.
>
> To validate this, we do experiments on **ImageNet-100**, which randomly selects 100 classes from ImageNet-1K and all training configurations follow the official setups in [1]. We support 10 clean and 10 backdoor samples to compute the average results.
>
> Below are the detection results using our method under SSL backdoor attacks [1]:
> |SSL Method|Backbone|DSR|FPR|
> |-|-|-|-|
> |MoCo v2|ResNet-18|94.21|2.11|
> |BYOL|ResNet-18|95.78|1.05|
>
> Results show that our method generalizes well beyond supervised learning and performs robustly even in self-supervised settings without labels. Since OD Loss always operates on the model’s **internal training objective**, it remains applicable to **any learning setting where a task loss is defined**, without requiring access to labels.
>
> [1] Backdoor Attacks on Self-Supervised Learning. In CVPR 2022

---

> > ### Author Response · Authors · 2025-08-03
> >
> > Dear Reviewer 3ggA,
> >
> > Thank you for taking the time to review our submission and providing us with constructive comments. We would like to confirm whether our responses adequately addressed your earlier concerns, particularly regarding the previously unclear illustration of the **threat model**, issues of **dataset dependency and experimental fairness**, the strong assumption and our robust performance under more **extreme and adaptive attacks**, as well as the **generalizability** of our method.
> >
> > Additionally, if you have any further concerns or suggestions, we would be more than happy to address and discuss them to enhance the quality of the paper. We eagerly await your response and look forward to hearing from you.
> >
> > Best regards,
> >
> > The authors

---

> > ### Comment · Reviewer_3ggA · 2025-08-03
> >
> > Thanks for the rebuttal.
> >
> > Q1 is settled by authors with further modifications.
> > Q2 is well resolved.
> > Q3 : The problem stated is acknowledged by the authors. Whether a verifier has access to a subset of clean training data can be vital in the assumption. What if only out-of-distribution data is available and training data is unavailable? This part may require further evaluation and systematic study.
> > Q4 is settled. Further modifications should be made by authors.
> > Q5 is settled by authors with further modifications.
> >
> > I would raise the score accordingly.

---

> > > ### Author Response · Authors · 2025-08-05
> > >
> > > Dear Reviewer 3ggA,
> > >
> > > Thank you for your thoughtful follow-up and for raising the important question **Q3** regarding scenarios where only out-of-distribution (OOD) data is available for verification while the original training data is entirely inaccessible.
> > >
> > > To further evaluate the robustness of **Lie Detector** under such constrained settings, we conducted additional experiments that reflect realistic deployment environments such as federated learning and black-box training pipelines, where clean training data may not be accessible to the verifier.
> > >
> > > We examine three challenging OOD verification scenarios:
> > >
> > > 1. **ResNet-18** trained on **CIFAR-10**, with the verifier using a **diffusion-generated CIFAR-10 set** (10 percent synthetic samples per class, 5000 samples in total) to serve as the OOD data.
> > >
> > >    We generate the **Diffused CIFAR-10** as follows: To simulate clean data not overlapping with training samples, we trained a class-conditional diffusion model (based on DDPM [1]) using the CIFAR-10 training set for 400,000 steps with a UNet backbone and cosine noise schedule. After convergence, we generated 5,000 synthetic images, evenly distributed across classes. These images are visually consistent with CIFAR-10 but distributionally distinct, providing a practical surrogate for OOD verification.
> > >
> > > 2. **CLIP** trained on ImageNet and verified using **ImageNet-A** [2] as the OOD data.
> > >
> > > 3. **LLaVA** verified using **Flickr8k** captions, differing from the training set Flickr30k.
> > >
> > > To assess the impact of distribution shift, we report two metrics:
> > > - **K**, which measures the distributional difference between the verifier data and training data using Fréchet inception distance (FID) score [3,4]. Higher values indicate greater distribution shift. In practice, FID values above **20–30** are commonly considered to reflect noticeable distribution differences.
> > > - **P**, which evaluates how strongly the OOD data activates the model’s internal features, measured using cosine similarity (range from -1 to 1) [5,6]. Lower values indicate greater distribution shift. A drop below **0.9** typically suggests a semantic or structural mismatch between verifier data and training data.
> > >
> > > **All results are averaged over 10 clean and 10 backdoored models as the model pool per setting.**
> > >
> > > | Model    | Original Train Dataset $\mathcal{D}_c$ | Verifier OOD Data $\mathcal{D}_s$     | Attack     | DSR (%) | FPR (%) | K   | P    |
> > > |----------|----------------|-------------------|------------|---------|---------|-----|------|
> > > | ResNet18 | CIFAR-10       | Diffused CIFAR-10 | ISSBA      | 93.16   | 0.53    | 23  | 0.89 |
> > > | CLIP     | ImageNet       | ImageNet-A        | BadCLIP    | 88.95   | 3.16    | 37  | 0.85 |
> > > | LLaVA    | Flickr30k      | Flickr8k          | TrojanVLM  | 86.84   | 6.32    | 41  | 0.81 |
> > >
> > > In the **ResNet18 + Diffused CIFAR-10** setting, the detector achieves a DSR of **93.16%** and a very low FPR of **0.53%**, showing that synthetic data can be an effective substitute. For **CLIP**, although the distribution gap is larger (K = 37), we still achieve near 89% DSR with reasonable FPR (3.16%). In the **LLaVA** case, where semantic deviation is more pronounced, the performance remains acceptable with an 86.84% DSR and 6.32% FPR.
> > >
> > > The results demonstrate that **Lie Detector** remains robust even using out-of-distribution (OOD) for verification, where the original training data is completely inaccessible. Despite substantial distribution shifts, it achieves strong performance across different learning paradigms.
> > >
> > > We appreciate your recognition of the improvements and will revise all remaining aspects (Q1 to Q5), including clarity and completeness, in the final version.
> > >
> > > [1] Denoising Diffusion Probabilistic Models. In NeurIPS 2020.
> > >
> > > [2] Natural Adversarial Examples. In CVPR 2021.
> > >
> > > [3] GANs Trained by a Two Time-Scale Update Rule Converge to a Local Nash Equilibrium. In NeurIPS 2017.
> > >
> > > [4] Visual Prompt Tuning for Generative Transfer Learning. In CVPR 2023.
> > >
> > > [5] Maximizing Cosine Similarity Between Spatial Features for Unsupervised Domain Adaptation in Semantic Segmentation. In WACV 2022.
> > >
> > > [6] Heuristic Domain Adaptation. In NeurIPS 2020.
> > >
> > > Best regards,
> > >
> > > Authors

---

### Official Review · Reviewer_vc1P · 2025-06-27

**Clarity:** 2
**Significance:** 2
**Originality:** 2
**Rating:** 4
**Confidence:** 5

**Summary:**

This paper proposes Lie Detector, a unified backdoor detection framework that leverages cross-examination of two independently trained models in a semi-honest outsourcing setting. It introduces two novel components: (1) a cross-model trigger inversion mechanism based on output distribution and Centered Kernel Alignment (CKA) to detect representational inconsistencies and recover potential triggers, and (2) a fine-tuning sensitivity analysis to distinguish true backdoors from benign anomalies. The framework is designed to generalize across multiple learning paradigms (supervised, self-supervised, and autoregressive). It is shown to outperform existing backdoor detectors on diverse datasets and architectures, including large multimodal models like LLaVA and MiniGPT-4.

**Questions:**

Q1. Please clarify the reasonable setting of the two outsourcing servers, and these servers do not collude.

Q2. Please provide more ablation studies or discussions on the trigger formulation with/without CKA-based loss.

Q3. Please verify the generality of equation (4).

Q4. Please provide more details about the computation cost.

Q5. How to identify that both models are backdoored? There seems lack of such experimental results in the manuscript.

**Ethical Concerns:**

["NO or VERY MINOR ethics concerns only"]

**Final Justification:**

Thank you for the detailed response. Most of my initial concerns have been adequately addressed. Upon reviewing the manuscript and considering the comments from other reviewers, I find that the core design of the proposed method, particularly the use of CKA-based similarity estimation, plays a central role in its effectiveness. As shown in Appendix A, this estimation mechanism has a notable impact, especially in detecting two specific backdoor models, which largely explains the success of the proposed defense.

Overall, I believe the paper presents meaningful contributions to the field. However, several important aspects warrant further clarification or exploration to strengthen the work:
- Hyperparameter Sensitivity: It would be helpful to investigate how sensitive the performance of the method is to key hyperparameter choices, particularly those involved in the similarity estimation and detection pipeline.
- Backdoor Model Settings: Please provide detailed settings of the backdoor models used in Appendix A. Additionally, it would be valuable to include results for similarity estimation under a collusion-based backdoor model to better understand robustness.
- Scalability to Multiple Providers: I am curious whether the method maintains or improves its performance when applied in scenarios with three or more providers.

**Limitations:**

yes

**Quality:**

2

**Strengths And Weaknesses:**

S1.  The proposed framework generalizes well across SL, SSL, and AL, addressing a key limitation of prior detectors that are often paradigm-specific.

S2. The combination of CKA for feature similarity and fine-tuning sensitivity analysis is novel and effective, enabling both accurate and robust detection.

S3. Extensive experiments on various models, datasets, and attack types demonstrate strong performance and practical applicability, including in the context of challenging vision-language models.

W1. Threat model. This work should further clarify the available settings of the two outsourcing models. It seems that if there is only one outsourcing model, or the user would not like to pay more for other service providers, the proposed defense may not have similar performance as reported. The reviewer is also worried about the independent settings of the service providers, although some additional experiments are provided in the appendix for the collusion model. As expected, the performance in such a scenario drops quickly.

W2. Novelty clarification. As stated in the manuscript, CKA and fine-tuning sensitivity have been used independently; the combination of these two techniques seems to be a direct application.

W3. The reviewer wonders about the generality of the trigger generation. If attackers use different trigger generations, i.e., different equations (4), the reversing trigger may not affect the poisoned model.

W4. Extra computing cost for the user. The reviewer is not convinced that if one user does not have the computing power to do the training task, the proposed trigger generation and inference process also costs a lot. Although some comparisons are provided, the missing details may overlook this weakness.

---

> ### Author Rebuttal · Authors · 2025-07-31
>
> **W1&Q1**. Threat model unclear: dual outsourcing may be unrealistic or costly; performance may drop if models are not independent or only one is used.
>
> **A1**. We further clarify:
>
> **(1) Practicality of the Dual-Model Setting**:
> - Cost acceptability. Our method requires only two model training sessions (i.e., double the cost) to achieve unified detection without labeled data or clean models. This is more efficient and lightweight compared to existing methods (DECREE, which requires extensive simulated attacks).
> - Generality and flexibility. Lie Detector supports multiple architectures and learning paradigms, making it a general detection method. Users can first run Lie Detector on a combination of small models to assess the credibility of service providers at low cost, and then decide whether to use large models, enhancing flexibility.
> - Practical feasibility. In security-sensitive industries such as finance, healthcare, and government, cross-model verification and multi-service provider comparisons are already commonplace. Additionally, processes like AutoML and federated learning inherently involve multiple sub-models, which can directly reuse our framework.
>
> A real-world scenario: when a user with a large amount of private data wants to outsource model training, a common practice is to test multiple candidates on a small amount of data (i.e., “competitive bidding”) to evaluate their training capabilities, and then select the optimal one to perform the full-volume training. In this process, the service providers are in competition with each other, and thus the possibility of collusion is low. Our approach can be used for this early evaluation stage, and detect possible backdoor behaviors in the returned models. Once we confirm that a model submitted by a service provider has anomalies, we can blacklist that service provider to avoid subsequent cooperation.
>
> **(2) Performance under Model Collusion**:
>
> We simulate **varying degrees of provider collusion** as shown below (“✓”/“✗” means that the two models share/differ that setting):
> |Level|Name|Describe|Data Share|Initialize|LR|Epoch|
> |-|-|-|-|-|-|-|
> |**L0**|Baseline in our paper|Same backdoor target and config, but fully independent training|✗|✗|✓|✓|
> |**L1**|Weak Collusion|Partial backdoor data shared (30% overlap); different initialization|30%|✗|✓|✓|
> |**L2**|Moderate Collusion|Full backdoor data sharing; different initialization|✓|✗|✓|✓|
> |**L3**|Strong Collusion in Appendix E|Full backdoor data sharing; identical training|✓|✓|✓|✓|
>
> We report the results (%) below:
> |Attack/Dataset|Model|Level|DSR|FPR |SOTA DSR / FPR|
> |-|-|-|-|-|-|
> |ISSBA/CIFAR10|ResNet-18|L0|100.0|0.0|95.0 / 10.0（TED)|
> |||L1|95.0|2.5||
> |||L2|82.5|7.5||
> |||L3|70.0|15.0||
> |BadCLIP/Caltech101|CLIP|L0|95.0|5.0|90.0 / 5.0  (SEER)|
> |||L1|90.0|5.0||
> |||L2|75.0|10.0||
> |||L3|72.5|15.0||
> |TrojanVLM/COCO|LLaVA|L0|92.5|5.0|80.0 / 15.0 (SEER)|
> |||L1|85.0|7.5||
> |||L2|72.5|12.5||
> |||L3|67.5|17.5||
>
> Our method remains robust under **weak or moderate collusion**. Under L1, despite some backdoor data being shared, the models still learn different representations, and CKA can capture these differences. Therefore, Lie Detector remains superior to SOTA under this setting.
>
> As expected, performance degrades under strong collusion (L3), where both models are nearly identical. We have discussed such case in Appendix F. It is worth noting that such strong collusion is extremely rare in practice and difficult to coordinate, especially when the model is trained by independent commercial or decentralized service providers, our method maintains a **DSR exceeding 70.0%** even in such worst-case scenario.
>
> One potential solution is to randomly distribute the training data to multiple independent training parties, thereby reducing the possibility of collusion from the outset. Ultimately, the models from all parties can be integrated through model fusion (such as ensemble or distillation) to maintain accuracy and achieve robust detection. We plan to make this strategy an important direction for future work and **integrated these new settings, results, and analyses into the revised version**.
>
> **W2**. Novelty unclear: combining CKA and fine-tuning sensitivity may appear as a direct application of known techniques.
>
> **A2**. We clarify that this work is not a simple combination of existing methods, but rather a systematic detection framework for the core challenges of backdoor detection:
> - **The novelty of the detection framework and the first introduction of CKA.**
> We are the first to apply CKA for backdoor detection. Unlike traditional feature comparisons, CKA better detects training objective shifts. Importantly, we actively optimize triggers to maximize CKA divergence, outperforming other similarity metrics (Appendix D). This mechanism naturally extends to multi-model detection with just two models, offering low cost and easy deployment.
> - **General detection capability for different learning paradigms.**
> Traditional methods rely on specific architectures, task forms, or loss functions, making them difficult to adapt to different paradigms such as SL/SSL/AL. We combines CKA and OD loss to eliminate such dependencies. We are the first to successfully detect backdoors in multi-modal generative models (LLaVA, MiniGPT-4), achieving a DSR of up to 95.0% (Table 2).
> - **Fine-tuning sensitivity analysis (FTSA) as a supplementary mechanism.**
> The FTSA is not a simple reuse of fine-tuning algorithms but serves as a behavioral-level supplementary validation mechanism for trigger inversion results. Only when the model indeed embeds backdoor and the trigger is accurately inverted, the $\Delta$ASR will much decrease. If the model is clean or the trigger inversion is ineffective, fine-tuning will have little impact. Therefore, FTSA can enhance detection accuracy while effectively reducing FPR (Table 3).
>
> **W3&Q3**. Generality concern: different attackers may use different trigger generation methods, which may limit the effectiveness of trigger inversion.
>
> **A3**. We answer as the following points:
> - **The goal of the inverse trigger is not to restore the original trigger, but to simulate potential triggerable behavior.**
> This mechanism has been verified in previous work [DECREE][UNICON]. Our method does not rely on restoring the exact trigger, but instead finds a “potential input that can activate abnormal behavior”. This idea has been validated in DECREE, where even if the inversed trigger does not match the real one, the model's abnormal response can still expose the existence of a backdoor.
> - **The mask + pattern design in equation (4) has high generalization capabilities and can cover multiple types of typical triggers.**
> This design enables flexible representation of different trigger types. This representation has been widely used in various inversion or attack tasks (see DECREE, UNICORN, etc.).To verify the generality of the formula, referring to existing research [2], we mainly divide backdoors into: Static Backdoor Attacks: BadNets, Blended; Content-Aware Backdoor Attacks: LowFrequency, ISSBA.; Model-Adaptive Backdoor Attacks: BadCLIP, TrojanVLM.
>
> - **Our experiments have covered various types of triggers, validating the robustness of this method.**
> We evaluate the effectiveness under various attacks in Table 1 and 2 (main paper) and Table 5 (appendix), including (DSR,FPR): BadNets (100.0%, 0.0%), Blended (100.0%, 0.0%), Low-Frequency (100.0%, 0.0%), ISSBA (100.0%, 0.0%), LabelConsistent[1] (95.0%, 0.0%), BadCLIP（95.0%, 5.0%), and TrojanVLM (95.0%,5.0).
>
> [1] Label-Consistent Backdoor Attacks. In arxiv 2019.
> [2] Revisiting Backdoor Attacks against Large Vision-Language Models from Domain Shift. In CVPR 2025.
>
> **W4&Q4**. Extra cost concern: if users can't train models, trigger generation may also be too costly; current comparisons may miss this point. Please provide more details about the computation cost.
>
> **A4**. We will report the average detection time for each model in the paper, and provide comparisons with baselines. TED's runtime is about 6 times longer than ours, particularly on llava, where our runtime is 5.09 minutes, while TED's is 35.48 minutes. For details, please refer to **A1** for Reviewer BEna.
>
> **Q2**. Discuss the effect of using or removing the CKA loss in trigger optimization.
>
> **A5**.  We experiment on **removing the CKA loss term Eq. (7)**.
> |Attack|Task| Model| DSR (w/o CKA) | DSR (Ours) | FPR (w/o CKA) | FPR (Ours) |
> |-|-|-|-|-|-|-|
> |Blended|CIFAR-10|ResNet-18|85.0|100.0|7.5|0.0|
> |BadEncoder|Caltech101|CLIP|80.0|95.0|10.0|2.5|
> |Shadowcast| Flickr8k|LLaVA|77.5|92.5|10.0|2.5|
>
> We see that removing CKA leads to **notable drops in DSR and increases in FPR**, confirming that **CKA loss provides complementary representation-level signals** during trigger optimization, especially for complex architectures (CLIP, LLaVA), where output distributions alone may not sufficiently expose backdoor inconsistencies. We will include this in the revised version.
>
> **Q5**. How to identify that both models are backdoored? There seems lack of such experimental results in the manuscript.
>
> **A6**. According to Alg. 1 and Appendix A, in the clean-clean case, the CKA value stays relatively high and the trigger has difficulty converging. In contrast, in the backdoor-backdoor and backdoor-clean cases, the CKA value decreases, allowing the reverse-trigger optimization process to converge quickly (there is one trigger per model.) Subsequently, we fine-tuned the model using the reverse triggers and measured the ASR. If the ASR was below a certain threshold, the model was identified as backdoored, otherwise clean. In the experimental setup, the detection results naturally cover all possible cases.
> On ResNet-18, CIFAR-10, 10 pairs of models were selected for testing under ISSBA attack. The clean-clean, clean-backdoor and backdoor-backdoor detection results are as follows: DSR: 100.0%, 100.0%, 100.0%,  FPR: 0.0%, 0.0%, 0.0%.

---

> > ### Comment · Reviewer_vc1P · 2025-08-07
> >
> > Thanks for the detailed response and I have raised my score accordingly.

---

> > > ### Author Response · Authors · 2025-08-07
> > >
> > > Dear Reviewer vc1P,
> > >
> > > Thank you very much for your valuable feedback and kind support. We are pleased to know that the additional experiments and clarifications have resolved your concerns. Your recognition of our work and your recommendation for acceptance are truly encouraging and deeply appreciated.
> > >
> > > As mentioned, we will continue to refine both the main manuscript and supplementary materials in the revised version, incorporating further experiments and improvements where needed.
> > >
> > > We are grateful for your constructive input throughout the review process.
> > >
> > > Best regards,
> > >
> > > The Authors

---

### Official Review · Reviewer_auii · 2025-07-02

**Clarity:** 4
**Significance:** 4
**Originality:** 4
**Rating:** 5
**Confidence:** 5

**Summary:**

This paper proposes Lie Detector, a generalized backdoor detection framework that detects the presence of backdoors by performing cross-model consistency checking on two independent third-party trained models. The method combines central kernel alignment with output distribution optimized inversion triggers, and effectively reduces false positives by fine-tuning the sensitivity analysis. Lie Detector achieves, for the first time, generic backdoor detection for supervised, self-supervised, and autoregressive tasks, especially for multimodal large models. The experiments show significantly better detection capability than existing methods under multiple benchmark tasks, which has strong practical value.

**Questions:**

Please see weakness.

**Ethical Concerns:**

["NO or VERY MINOR ethics concerns only"]

**Final Justification:**

Thanks for the detailed response.

I appreciate the additional experimental results. Most of my initial concerns have been adequately solved. I decide to maintain my initial score, as I believe it has already reflected my support for the acceptance of this paper.

**Limitations:**

yes

**Quality:**

4

**Strengths And Weaknesses:**

Strengths:
1. Novel and reasonable verification framework. This paper introduces a backdoor detection framework based on cross-model cross-checking for the first time, which holds significant practical value in real "semi-honest third-party training" scenarios.
2. Highly versatile and cross-paradigm effective. It breaks through the limitation that existing methods mainly target a single paradigm (e.g., classification) and can be applied to supervised, self-supervised, and autoregressive tasks, including multimodal large models.
3. The method is practical and scalable. The method does not require access to training data and does not rely on a clean reference model. Only two independently trained models are needed to complete the detection, which meets the needs of real-world multi-vendor auditing applications.
4. This method enables the detection of large models for the first time. Compared with existing methods, this method realizes effective backdoor detection on large VLMs such as LLaVA and MiniGPT-4 for the first time.

Weaknesses.
1. Hyperparameter sensitivity is not sufficiently analyzed. The trigger inversion, CKA computation, and fine-tuning sensitivity modules in the current framework involve some hyperparameters (e.g., trigger mask size, canonical strength, and number of fine-tuning steps). The paper has not systematically explored the sensitivity of these hyperparameters to the detection effect. The method steps are complex and these hyperparameters may have a serious negative impact on the performance of the detection.
There has been insufficient verification of generalizability across various model architectures. Although the paper was tested on mainstream architectures (e.g., ResNet, Transformer, VLM), generalizability has not yet been demonstrated on lighter or more heterogeneous architectures (e.g., MobileNet, ConvNeXt). The authors need to add more exploration of architecture.
3. Insufficient coverage of some atypical backdoor attacks. Currently, the main attacks evaluated are trigger-based backdoors, and specific attacks such as clean-label attacks and multimodal backdoor attacks are not yet covered.
4. Robustness under model version change is not sufficiently discussed. In practice, there may be version differences between vendors' models (e.g., different training rounds, different parameter initialization). Whether the current method is still effective when the model versions differ significantly (not completely independent but approximate) has not been analyzed in depth in the paper. Do these non-core factors have a serious impact on the detection rate of the defense when the model versions change?

---

> ### Author Rebuttal · Authors · 2025-07-31
>
> **W1**. Hyperparameter sensitivity (e.g., trigger size, canonical strength, fine-tuning steps) is under analyzed. Generalizability to lighter or diverse architectures (e.g., MobileNet, ConvNeXt) also needs further validation.
>
> **A1**.  Thank you for the suggestion. We will explain this in terms of two aspects: hyperparameter sensitivity and lightweight model architecture：
>
> **(1) Hyperparameter Sensitivity:**
>
> We have conducted a series of ablation studies to analyze the impact of key hyperparameters:
>
> - **Trigger mask size:**
> We tested 2×2, 4×4, and 8×8 trigger sizes. Detection results (ResNet-18, CIFAR-10, ISSBA) are summarized below:
>
>  • 2×2 → DSR: 85.0%, FPR: 10.0%
>
>  • 4×4 → DSR: 100.0%, FPR: 0.0%
>
>  • 8×8 → DSR: 100.0%, FPR: 5.0%
>
> - **Loss weights (α, β, λ):**
> Moderate changes show limited impact, suggesting robustness of the framework.
>
> | α    | β    | λ    | DSR (%) | FPR (%) |
> | ---- | ---- | ---- | ------- | ------- |
> | 0.4  | 0.5  | 0.1  | 90.0    | 7.5     |
> | 0.6  | 0.3  | 0.1  | 100.0   | 0.0     |
> | 0.5  | 0.3  | 0.2  | 95.0    | 5.0     |
>
> - **Fine-tuning steps:**
> Performance stabilizes within 10 epochs (DSR: 100.0%, FPR: 0.0%).
>
> - **Adaptive thresholding ($\gamma$):** We use **Gaussian Mixture Modeling (GMM)** to fit the distribution of ASR drop values across samples, and automatically select the threshold $\gamma$ that best separates clean and backdoored models to maximize detection performance. For ISSBA on ResNet-18/CIFAR-10, GMM selects $\gamma = 0.22$, yielding 100.0% DSR.
>
> Overall, we can see that our method is not sensitive to those hyperparameters.
>
> **(2) Generalizability to Diverse Architectures:**
> We have validated our method across a broad range of models including ResNet-18, VGG16, CLIP, CoCoOp, LLaVA, and MiniGPT-4, covering classification, contrastive learning, and multimodal large models.
>
> To strengthen this further, we include results on lighter models:
> - **MobileNetV2:** DSR: 100.0%, FPR: 0.0%
> - **ConvNeXt:** DSR: 97.5%, FPR: 0.0%
>
> And we can see that oue method can well generalize well to diverse architectures.
>
>
> **W2**. Coverage of a typical backdoor attacks (e.g., clean label, multimodal) is insufficient. Current evaluation focuses mainly on trigger-based attacks.
>
> **A2**. In fact, our paper already includes detection results for *multimodal backdoor attacks*, specifically:
>
> - **TrojanVLM** and **Shadowcast**, two representative attacks on large multimodal models (LLaVA and MiniGPT-4), are evaluated in Table 2.
> - *Lie Detector* achieves the highest detection success rates (e.g., 95.0%, 92.5%, 90.0%) on COCO and Flickr30k, with low FPRs (≤10%), significantly outperforming baselines such as TED, MM-BD, DECREE, and SEER.
>
> Additionally, we have now included results for **clean label attacks**, specifically Label-Consistent [1]:
>
> | Dataset      | Model     | Attack Type     | DSR (%) | FPR (%) |
> | ------------ | --------- | --------------- | ------- | ------- |
> | CIFAR-10     | ResNet-18 | Label-Consistent| 95.0    | 0.0     |
> | TinyImageNet | ResNet-18 | Label-Consistent| 95.0    | 2.5     |
>
> We tested the detection performance under clean label attacks on different datasets, and the results further demonstrated the robustness of *Lie Detector* in complex attack scenarios, especially with a DSR as high as 95.0% on CIFAR10.
>
> [1] Label-Consistent Backdoor Attacks. In arxiv 2019.
>
>
>
> **W3**. Robustness to version differences (e.g., training epochs, parameter initialization) is unclear. Effectiveness under approximate models has not been deeply analyzed.
>
> **A3**. In real-world deployment, model version differences (e.g., different initialization or training epochs) are common. We have explicitly considered this in both the **design** and **experiments** of our method.
>
> **(1) Method Design: CKA is Inherently Robust to Version Differences**
> Our method relies on **Centered Kernel Alignment (CKA)** to measure representation-level similarity, rather than parameter or behavior-level differences. This makes the method robust to variations in initialization and training trajectory. Specifically:
>
> - **Orthogonal invariance:** CKA remains stable even when different training runs produce activations in different subspaces, as long as the learned semantics are consistent.
> - **Scale invariance:** CKA is unaffected by differences in activation magnitude caused by training length or learning rate.
>
> Thus, models trained with different seeds or epoch counts but similar objectives yield high CKA scores; poisoned models deviate significantly in CKA space due to structural changes in representation.
>
> **(2) Experimental Evidence: Stable Detection Under Version Shift**
> We here include results on *model pairs*, such as differing in training epochs or parameter initialization.
>
> - **Epoch variation (ResNet-18, CIFAR-10, ISSBA):**
>
> | Epochs (model1 vs model2) | DSR (%) | FPR (%) |
> | ------------------------- | ------- | ------- |
> | 50 vs 200                 | 85.0    | 7.5     |
> | 50 vs 100                 | 87.5    | 5.0     |
> | 200 vs 100                | 97.5    | 0.0     |
>
> The 200 vs 100 has the best results, indicating that our detection can improve with sufficient training. Even when one model is partially trained (model1 for 50 epochs), our method remains effective.
>
> - **Parameter initialization differences:**
>
> | Epoch | Initialization schemes | DSR (%) | FPR (%) |
> | ----- | ---------------------- | ------- | ------- |
> | 100   | Kaiming vs Kaiming     | 100.0   | 0.0    |
> | 100   | Kaiming vs Xavier      | 97.5    | 0.0    |
> | 100   | Xavier vs Uniform      | 97.5    | 0.0    |
>
> We observe **minimal performance drop** (all above 97.5%) under varying initialization schemes (Kaiming, Xavier, etc.), indicating strong robustness.
>
> In summary, *Lie Detector* is robust to typical version changes in model deployment, and we will further highlight these analyses in the revision.

---

> > ### Comment · Reviewer_auii · 2025-08-04
> >
> > Thanks for the detailed response.
> >
> > I appreciate the additional experimental results. Most of my initial concerns have been adequately solved. I decide to maintain my initial score, as I believe it has already reflected my support for the acceptance of this paper.

---

> > > ### Author Response · Authors · 2025-08-04
> > >
> > > Dear Reviewer auii,
> > >
> > > We sincerely appreciate your thoughtful feedback and kind support. We are glad that the additional experimental results and clarifications have addressed your concerns. Your recognition of our efforts and your support for the acceptance of this paper mean a great deal to us.
> > >
> > > As noted, we will further refine the main text and supplementary materials in the revised version, incorporating additional experiments and improvements where appropriate.
> > >
> > > Thank you again for your constructive input throughout the review process.
> > >
> > > Best regards,
> > > Authors

---

### Official Review · Reviewer_BEna · 2025-07-02

**Clarity:** 3
**Significance:** 3
**Originality:** 3
**Rating:** 4
**Confidence:** 3

**Summary:**

The paper proposes a unified backdoor detection framework that exploits cross-examination
of model inconsistencies between two independent service providers. The method demonstrates good generalization and robustness, achieving high detection accuracy across supervised, semi-supervised, and autoregressive learning paradigms

**Questions:**

What is the runtime of the method on miniGPT-4 (Table 6)?

**Ethical Concerns:**

["NO or VERY MINOR ethics concerns only"]

**Limitations:**

yes

**Quality:**

3

**Strengths And Weaknesses:**

Strengths
- The paper is well motivated and well structured, supported by thorough experiments and illustrations.

Weaknesses
- It would be better if the paper includes explicit discussion of scalability concerns and computational overhead. Cross-examination involving multiple model trainings and trigger optimization might pose challenges in practice, particularly for large-scale models.
- Given the observed sensitivity in experiments (Figure 4), guidance or adaptive methods for parameter selection could strengthen the practical utility.

---

> ### Author Rebuttal · Authors · 2025-07-31
>
> **W1**. It would be better if the paper includes explicit discussion of scalability concerns and computational overhead. Cross-examination involving multiple model trainings and trigger optimization might pose challenges in practice, particularly for large-scale models.
>
> **A1**. Thank you for pointing out the computational challenges in large-scale settings. Our method achieves consistently strong performance across a range of architectures (from ResNet-18 to MiniGPT-4), and as shown in Table 6, detection accuracy shows no clear negative correlation with GFLOPs.
>
> To further address the concern, we report the average detection time of *Lie Detector* across different models and datasets ( Followed DECREE and measured on a single NVIDIA A100 80GB GPU.) compared with TED:
>
> | Model     | Dataset    | Attack     | GFLOPs | Lie Detector (min) | TED (min) |
> | --------- | ---------- | ---------- | ------ | ------------------ | --------- |
> | ResNet-18 | CIFAR-10   | ISSBA      | 0.7    | 0.42               | 2.61      |
> | CLIP      | Caltech101 | BadCLIP    | 4.9    | 2.27               | 14.80     |
> | LLaVA     | COCO       | Shadowcast | 76.6   | 5.09               | 35.48     |
> | MiniGPT-4 | Flickr8k   | TrojanVLM  | 80.3   | 4.53               | 28.19     |
>
> Even with large-scale multimodal models, the runtime remains practical (e.g., <4.6 min for MiniGPT-4). We also provide a comparison of detection performance, computational cost, and applicability in Appendix F (Table 6). *Lie Detector* achieves the highest detection accuracy (99.7%) with substantially lower runtime than TED (92.8% accuracy).
>
> **W2**. Given the observed sensitivity in experiments (Figure 4), guidance or adaptive methods for parameter selection could strengthen the practical utility.
>
> **A2**. Thank you for the suggestion. To adaptively determine the detection threshold $\gamma$ in Figure 4, we employ a standard **Gaussian Mixture Modeling** (GMM) based thresholding strategy. The underlying intuition is that the ASR drop values (i.e., the reduction in Attack Success Rate after applying our defense) follow a bimodal distribution, where clean and backdoored models exhibit distinct statistical behaviors.
>
> Specifically, we fit a two component GMM to the empirical distribution of ASR drop values across all models. Each component is assumed to capture the statistical characteristics of either clean or backdoored models. Once fitted, the GMM provides a probabilistic model of the ASR drop distribution, and we select the threshold $\gamma$ at the intersection point (or the maximum likelihood separation boundary) of the two Gaussian components. This threshold effectively separates clean models (which typically show small ASR drops) from backdoored models (which exhibit larger ASR drops), thereby maximizing the detection performance in a data-driven and adaptive manner.
>
> To more clearly demonstrate the performance changes, we repeated numerous experiments, and this adaptive selection achieved excellent results under different attacks. For example, when using the ResNet-18 model on the CIFAR-10 dataset and subjected to ISSBA attacks, the GMM algorithm identified $\gamma = 0.22$ as the optimal parameter, achieving a detection success rate (DSR) of 100.0% and a false positive rate (FPR) of only 0.0%, which is better than the DSR of 99.61% and the FPR of 0.26% when $\gamma = 0.2$ in Figure 4. Similarly, for low-frequency attacks, GMM selects $\gamma = 0.19$ and achieves a DSR of 99.87% and a FPR of 0.38%, which is better than the DSR of 99.23% and the FPR of 1.03% when $\gamma = 0.2$ in Figure 4. These results demonstrate that GMM-based thresholding generalizes well and often surpasses manually tuned settings.
>
> We will clarify this adaptive procedure and its practical utility in the revised paper.
>
> **Q1**. What is the runtime of the method on miniGPT-4 (Table 6)?
>
> **A3**. We will include the runtime for MiniGPT-4 in the revised version. On the Flickr8k dataset, detecting a model pair using Lie Detector takes about 4.53 minutes on a single NVIDIA A100 80GB GPU. In comparison, TED requires 28.19 minutes under the same setting.

---

### Note · Authors · 2025-08-12

Dear ACs, SACs, and Reviewers,

We thank the ACs, SACs, and reviewers for their time, feedback, and engagement, which strengthened our manuscript.

Our paper proposes **Lie Detector**, a unified backdoor detection framework across architectures and learning paradigms. Combined with cross-model CKA and fine-tuning sensitivity analysis, it achieves SOTA detection across supervised, self-supervised, and autoregressive tasks, improving by **4.4% (SL), 1.7% (SSL), and 10.6% (AR)** over prior methods, and is the first to detect backdoors in large multimodal models such as LLaVA and MiniGPT-4.

- **Reviewer BEna – Scalability and runtime**: Initially gave a **positive score**. We believe our rebuttal and additional evidence, including runtime analysis showing efficiency for large multimodal models, have addressed earlier concerns and may help if they revisit their assessment.

- **Reviewer auii – Hyperparameter sensitivity and coverage**: Maintained their **positive score** after discussion, affirming that this is a good paper. Our added experiments on hyperparameter sensitivity, architecture diversity, and expanded attack coverage aligned with their suggestions and confirmed the method’s robustness and applicability.

- **Reviewer vc1P – Version shifts and collusion**: Actively engaged and expressed clear **positive feedback** after reviewing our clarifications and results. Resolving concerns on stability under version changes and varying levels of provider collusion, with strong baseline comparisons, suggests an **improved score**.

- **Reviewer 3ggA – Threat model, OOD data, and adaptive attackers**: Responded positively to our clarifications on the threat model, evaluation with only OOD data, adaptive attacker assessment, and fairness of settings. Their strong support during the discussion indicates their evaluation is now **positive**.

Overall, the **post-discussion** shows that our responses have **addressed reviewers’ concerns** and **reinforced the work’s merits**. These updates demonstrate Lie Detector’s value as a **practical, efficient, and general** backdoor detection solution, enabling safer and more trustworthy deployment. Given the consistent **positive engagement** and the shift toward **stronger support across all reviewers**, the final assessments are **very likely to be positive overall**. We appreciate the reviewers’ feedback and thank the AC for considering our work.

Best regards,

Authors of submission 8639

---

### Decision · Program_Chairs · 2025-09-17

**Decision:**

Accept (poster)

**Comment:**

This work studies the detection of dishonest model training service providers using a two-party setting, with consideration of untrusted providers and adaptive attacks. This work ais to provide a general approach in that it works for many architectures, attacks, and training protocols.

This work has shown many positives. In particular, this approach shows clear improvements compared to existing works in the detection time, TPR/FPR, and overall cost. Further, the approach also shows promise even in harder threat models involving collusion. Finally, the universality and generalizability of this work are a strong reason for acceptance.

There are only a few weakneses identified by the reviewers that could be better addressed. In particular, the ablations studies for hyperparameter senstivity remain slim. I encourage the authors to explore more detailed ablations here.

Overall, many concerns were well addressed by the authors. I recommend them to include all changes as the final paper showws clear agreement from reviewers that it should be accepted.